

# Embryonic development of the moon jellyfish *Aurelia aurita* (Cnidaria, Scyphozoa): another variant on the theme of invagination

Yulia Kraus[1,2], Boris Osadchenko[3] and Igor Kosevich[3]

[1] Department of Evolutionary Biology, Faculty of Biology, Lomonosov Moscow State University, Moscow, Russia
[2] Koltzov Institute of Developmental Biology of the Russian Academy of Sciences, Moscow, Russia
[3] Department of Invertebrate Zoology, Faculty of Biology, Lomonosov Moscow State University, Moscow, Russia

Corresponding author
Igor Kosevich, ikosevich@gmail.com

## ABSTRACT

**Background**. *Aurelia aurita* (Scyphozoa, Cnidaria) is an emblematic species of the jellyfish. Currently, it is an emerging model of Evo-Devo for studying evolution and molecular regulation of metazoans' complex life cycle, early development, and cell differentiation. For *Aurelia*, the genome was sequenced, the molecular cascades involved in the life cycle transitions were characterized, and embryogenesis was studied on the level of gross morphology. As a reliable representative of the class Scyphozoa, *Aurelia* can be used for comparative analysis of embryonic development within Cnidaria and between Cnidaria and Bilateria. One of the intriguing questions that can be posed is whether the invagination occurring during gastrulation of different cnidarians relies on the same cellular mechanisms. To answer this question, a detailed study of the cellular mechanisms underlying the early development of *Aurelia* is required.

**Methods**. We studied the embryogenesis of *A. aurita* using the modern methods of light microscopy, immunocytochemistry, confocal laser microscopy, scanning and transmission electron microscopy.

**Results**. In this article, we report a comprehensive study of the early development of *A. aurita* from the White Sea population. We described in detail the embryonic development of *A. aurita* from early cleavage up to the planula larva. We focused mainly on the cell morphogenetic movements underlying gastrulation. The dynamics of cell shape changes and cell behavior during invagination of the archenteron (future endoderm) were characterized. That allowed comparing the gastrulation by invagination in two cnidarian species—scyphozoan *A. aurita* and anthozoan *Nematostella vectensis*. We described the successive stages of blastopore closure and found that segregation of the germ layers in *A. aurita* is linked to the 'healing' of the blastopore lip. We followed the developmental origin of the planula body parts and characterized the planula cells' ultrastructure. We also found that the planula endoderm consists of three morphologically distinct compartments along the oral-aboral axis.

**Conclusions**. Epithelial invagination is a fundamental morphogenetic movement that is believed as highly conserved across metazoans. Our data on the cell shaping and behaviours driving invagination in *A. aurita* contribute to understanding of morphologically similar morphogenesis in different animals. By comparative analysis,
we clearly show that invagination may differ at the cellular level between cnidarian species belonging to different classes (Anthozoa and Scyphozoa). The number of cells involved in invagination, the dynamics of the shape of the archenteron cells, the stage of epithelial-mesenchymal transition that these cells can reach, and the fate of blastopore lip cells may vary greatly between species. These results help to gain insight into the evolution of morphogenesis within the Cnidaria and within Metazoa in general.

# INTRODUCTION

The moon jellyfish *Aurelia* (Lamarck, 1816) is an emblematic medusa. *Aurelia* belongs to the class Scyphozoa, which, together with the classes Hydrozoa, Cubozoa, and Staurozoa, comprises the medusozoan cnidarians (*Collins, 2002*; *Collins, 2009*). Like nearly all medusozoans, *Aurelia* has a complex life cycle. *Aurelia* medusa is a planktivore bearing complex neural and sensory system. It produces gametes developing into short-living planula larva metamorphosing into asexual polyp after the settlement. Polyp produces multiple juvenile medusae (ephyrae) in the course of asexual reproduction called strobilation. Recently, 28 species of *Aurelia* have been recognized mostly based on molecular data (*Collins, Jarms & Morandini, 2021*; *Dawson & Jacobs, 2001*; *Lawley et al., 2021*). Among them, *A. aurita* (Linnaeus, 1758) and *A. coerulea* (von Lendenfeld, 1884) (= *Aurelia sp. 1*; *Dawson & Jacobs, 2001*) are invasive species widely distributed around the globe including the Northwest Pacific and Atlantic coast of Europe.

Although *Aurelia* does not fulfill all requirements of a laboratory model object (it is problematic to explore the entire life history of the large jellyfish in the laboratory, the generation time is relatively long, *etc.*) (*Bolker, 1995*; *Darling et al., 2005*), it is an emerging model for investigation of certain questions of Evo-Devo. The *Aurelia* genome has been sequenced (*Aurelia sp.1*; *Gold et al., 2019*), and the molecule framework controlling the polyp-to-jellyfish transition have been uncovered (*A. aurita*; *Fuchs et al., 2014*). Current research on *Aurelia* focuses on many topics, including molecular regulation of complex life cycle, life cycle evolution, cell differentiation, development of the nervous system and sense organs (*Brekhman et al., 2015*; *D'Ambra et al., 2021*; *Gold et al., 2019*; *Gold et al., 2015*; *Khalturin et al., 2019*; *Liu et al., 2018*; *Nakanishi et al., 2015*; *Nakanishi, Hartenstein & Jacobs, 2009*; *Nakanishi et al., 2008*).

The embryogenesis of *Aurelia* is also very valuable for Evo-Devo investigations. As a reliable representative of the class Scyphozoa, *Aurelia* can be used for comparative analysis of developmental mechanisms within Cnidaria and between Cnidaria and Bilateria. A large number of papers and a long history of research on *Aurelia* show that its early development has been studied exhaustively. Indeed, it was intensively studied in the XIX-XX centuries (*e.g.*, *Claus, 1883*; *Goette, 1887*; *Hargitt & Hargitt, 1910*; *Hyde, 1894*; *Smith, 1891*; *Yuan et al., 2008*). It was finally summarized by *Hargitt & Hargitt (1910)* that

gastrulation in *Aurelia* proceeds via invagination of the presumptive endoderm. At the same time, the detailed description and analysis of *Aurelia* developmental stages, which precede planula metamorphosis, is still absent (*Yuan et al., 2008*). Within Cnidaria, gastrulation via invagination is known only for anthozoans and scyphozoans. Despite being representatives of the same phylum, anthozoans and scyphozoans are phylogenetically distant (*Khalturin et al., 2019*), so the question can be posed whether invagination in different cnidarians relies on the same cellular mechanisms. The detailed study on cellular mechanisms of gastrulation in anthozoans is available for the only species—the sea anemone *Nematostella vectensis* (*Fritzenwanker, Saina & Technau, 2004*; *Kraus & Technau, 2006*; *Magie, Daly & Martindale, 2007*; *Technau, 2020*).

In this article, we report a comprehensive study of the embryonic development of *Aurelia aurita* from the White Sea population based on the modern methods of microscopy. We describe developmental stages from early cleavage up to the planula larva focusing on the morphogenetic mechanisms underlying gastrulation. We characterize the dynamics of cell shape changes and cell behavior in the course of the endoderm invagination. Moreover, we compare invagination in two cnidarian species—*A. aurita* and *N. vectensis*—and clearly show that morphogenetic processes (*e.g.*, invagination) of different species, which look similar on the level of gross-morphology, might differ at the cellular level. The number of cells comprising the archenteron and involved in invagination, changes in cell shape, the degree to which archenteron cells undergo epithelial-mesenchymal transition, and the fate of blastopore lip cells were found to differ significantly between these species. Such a comparative approach helps to gain insight into evolutionary relationship of developmental pathways within cnidarians and into the evolution of morphogenesis in metazoans in general.

## MATERIAL AND METHODS

### Animals

The medusae of *Aurelia aurita* are diecious. The oocytes develop in gonads during an entire season of sexual reproduction (that is from the middle of June until the middle/end of August at the White Sea). After completion of maturation, the oocytes fall into the gastric cavity, and the ciliary beating transfers them to the brood pockets of the oral arms (*Claus, 1883*) (Fig. 1A). Very likely, fertilisation occurs during oocyte migration to the oral arms (*Hargitt & Hargitt, 1910*). Embryos develop in the oral arm pockets until the planula-larva stage (*Berrill, 1949*; *Ishii & Takagi, 2003*) (Fig. 1A).

Embryos at different stages of development were obtained from the female medusae collected near N.A. Pertzov White Sea Biological Station (Lomonosov Moscow State University) (66°33′07.3″N33°06′55.7″E). The embryos were sucked out from the brood pockets with a plastic pipette, placed to the glass bowl and washed several times with natural seawater to decrease the amount of mucus, and kept at 8–12 °C in the filtered sea water.

### Light and electron microscopy

Living embryos were observed and imaged under a stereo microscope Leica M165C equipped with a digital camera Leica DFC420 (5.0MP), and a dissecting microscope Leica

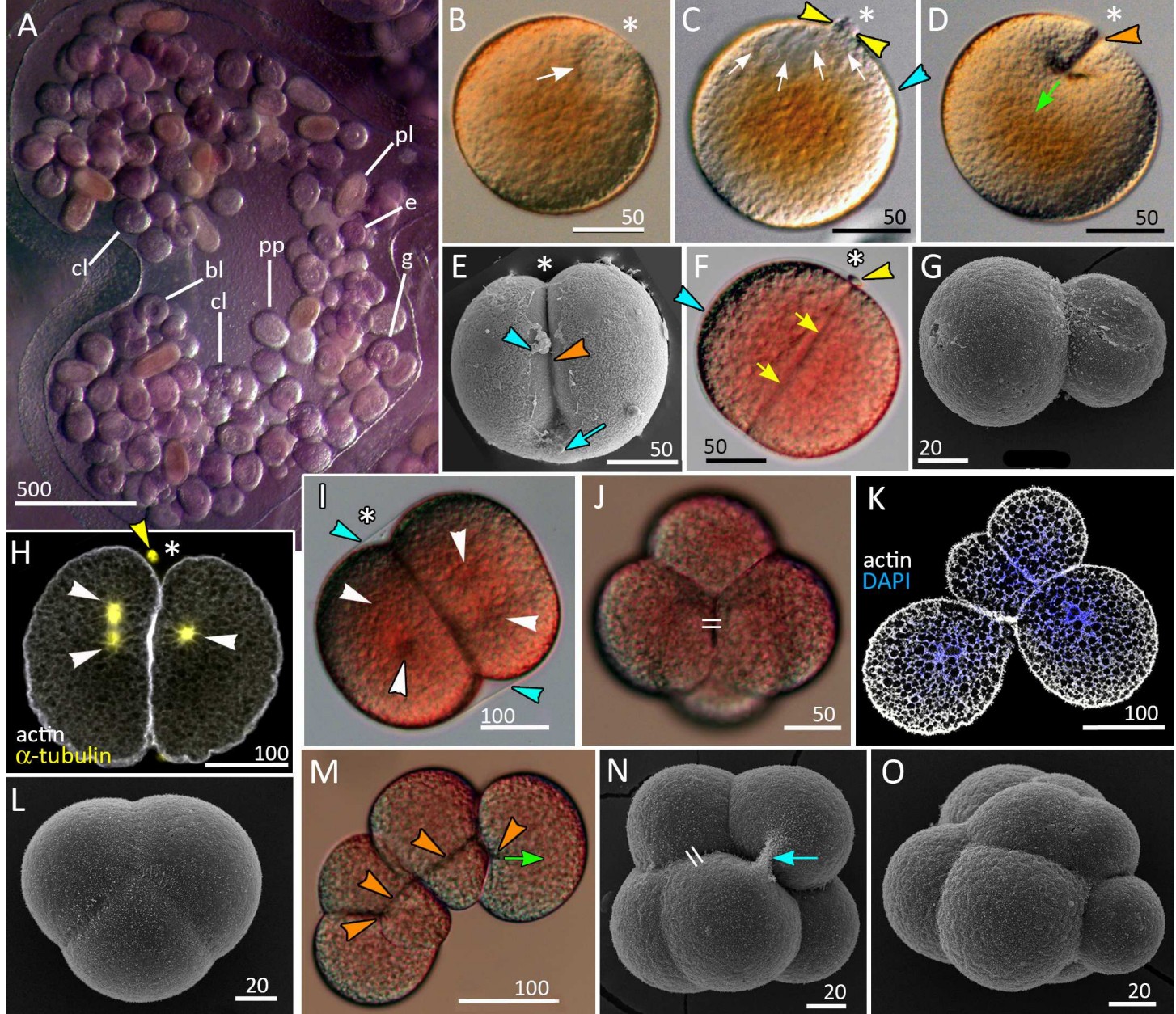

**Figure 1** **Brooding of eggs and embryos on female medusa (A) and early cleavage (B–O) of *Aurelia aurita*.** (A) A brood pocket in the medusa oral arm containing fertilized eggs and embryos at various stages of development. (B) Immature oocyte with large germinal vesicle (white arrow) near the animal pole. (C) Oocyte with two polar bodies (yellow arrowheads) in the perivitelline space; white arrows indicate the area occupied by the ruptured germinal vesicle. (D) The first cleavage furrow (orange arrowhead) originates at the animal pole (green arrow shows the direction of cleavage furrow spreading). (E) Cleavage furrow (orange arrowhead) spreads toward the vegetal pole; vitelline envelope ruptured during sample processing. (F) 2-cell stage, blastomeres are equal in size; yellow arrows indicate contact between two blastomeres. (G) Two-cell stage, unequal cleavage. (H, I) Embryos preparing for the second cleavage. Cleavage furrows not yet formed; white arrowheads indicate the visible poles of the mitotic spindles. (continued on next page...)

**Figure 1 (…continued)**
Note that the mitotic spindles are shifted to the animal pole in (H) and lie in the centers of the blastomeres in (I). (J) Four-cell stage, the blastomeres exhibit the compact packing, all blastomeres are the same size; the double stroke marks the contact between the nonsister blastomeres (*in vivo*). (K) Confocal section of an embryo at the four-cell stage showing unequal cleavage (*in vitro*). (L) Embryo with asynchronous cleavage consisting of three blastomeres (*in vivo*). (M, N) Third round of cleavage (*in vitro*). (M) Displacement of forming blastomeres proceeds simultaneously with the elongation of cleavage furrows. Blastomeres "twist" against each other to reach compact cell packing shown in (N); orange arrowheads—forming cleavage furrows, green arrow—spreading of the furrow. (N) The third cleavage is nearly complete, resulting in a tetrahedral arrangement of blastomeres with the contacts between the nonsister cells (double stroke). (O) Embryo with unequal and asynchronous cleavage consisting of 12 blastomeres (*in vivo*). Abbreviations: bl, blastula; cl, cleavage stage; e, fertilized egg; g, gastrula; p, immature planula; pp, preplanula. White asterisk marks the animal pole. Blue arrowhead indicates vitelline envelope. Yellow arrowhead points to the polar body. Blue arrow indicates the cytoplasmic bridge between two forming blastomeres. Light microscopy: A–D, F, I, J, M; SEM: E, G, L, N, O; CLSM: H, K.

DM2500 equipped with a digital camera Leica DFC420C (5.0MP) (Leica, Germany). The post-processing of the data and the projections with greater focal depth were made with LAS V.3.6.0 software (Leica).

For the light and electron microscopy samples were fixed overnight at 4 °C in 2.5% glutaraldehyde (GA, EM grade, Ted Pella, Inc., #18426) in phosphate buffer with addition of NaCl (pH 7.4; 0.83 Osmol; $NaH_2PO_4 \bullet H_2 0 – 1,8$ g, $Na_2HPO_4 \bullet 7H_2O – 23,25$ g, NaCl—5,0 g, $H_2 0$—up to 925 ml; approximately 0,1M concentration) (*Millonig, 1964*), and postfixed in 1% $OsO_4$ in the same buffer (1 h, room temperature, RT). After washing with the same buffer, the specimens were dehydrated through an ethanol and acetone, and embedded in Araldite502/Embed-812 embedding media (Electron Microscopy Sciences, Cat #13940). Semithin (0.5–1 μm thick) and ultrathin (60–80 nm) sections were cut with a Leica EM UC6 ultratome (Leica, Germany).

Semithin sections were stained with a mixture of toluidine blue and methylene blue (*Mironov, Komissarchik & Mironov, 1994*) and studied with a Leica DM5000 microscope equipped with a Leica DFC420C (5.0MP) digital camera (Leica, Germany).

Ultrathin sections for **transmission electron microscopy** (TEM) stained with uranyl acetate followed by lead nitrate were examined by the JEM-1011 JEOL and JEM-100 B-1 JEOL transmission electron microscopes (JEOL, Japan).

The samples for **scanning electron microscopy** (SEM) after the fixation and washing in phosphate buffer were dehydrated in ethanol series and acetone, critical point-dried in a HCP-2 Critical Point Dryer (Hitachi), mounted on stubs, sputter coated with platinum and palladium, and viewed in SEI mode at accelerating voltage 20 kV with scanning electron microscopes JSM-6380LA (JEOL, Japan; SEM Control User Interface Version 7.11 software) and Camscan-S2 (Cambridge Instruments, UK; MicroCapture software, SMA Ltd, Russia).

For **confocal laser microscopy** specimens were fixed in 4% paraformaldehyde (PFA; Fluka, Germany) in 0.1 M phosphate buffer (PBS; Fluka, Germany) overnight. After brief washing with PBS, embryos were incubated in a block solution (BS) with 1% bovine serum albumin (BSA; Sigma, St. Louis, MO, USA), 0.1% cold water fish skin gelatin (Sigma), 0.5% Triton-X100 (Ferak Berlin, Germany), and 0.05% Tween 20 (Sigma) for 24 h. Further on the samples were incubated at +4°C for 72 h in the 1:1 mixture of primary antibodies: mouse monoclonal anti-tyrosinated tubulin (1:2000; Sigma Cat # T9028) and mouse monoclonal anti-acetylated tubulin (1:2000; Sigma Cat #T6793). Then

the samples were washed four times for 3 h each in BS, and incubated for 72 h at 4 °C with a Donkey Anti-Rabbit IgG Antibodies labeled with Alexa Fluor 546 (1:500; Molecular Probes, #A10040).

After washing the samples at 4 °C in BS (four times for 2 h each), they were rinsed with PBS and stained for 1 h with a mixture of DAPI (100 ng/ml; Sigma) and BODIPY FL phallacidin (1:100; # B607; Molecular Probes). Following brief (20 min) washing in PBS the samples were mounted on a cover slip covered with poly-L-lysine (Sigma-Aldrich, St. Louis, MO, USA), and then cleared and mounted in Murray Clear (a 2:1 mixture of benzyl benzoate and benzyl alcohol) (*Von Dassow, 2010*).

The samples were studied using a Nikon A1 confocal microscope (Tokyo, Japan). Z-projections were generated using NIS-Elements D4.50.00 (Nikon) and Image J V.1.43 (https://imagej.nih.gov/ij/) and processed with Adobe Photoshop CS5 Extended v. 12.0.3 ×32 (Adobe Systems, San Jose, CA, USA).

Negative controls included specimens processed without incubation in primary antibodies. The autofluorescence control was prepared without addition of fluorochrome (secondary antibodies). Negative controls revealed no unspecific labelling.

## RESULTS

The development of germ cells occurs in 'gonads' located at the bottom of the gastric pouches of the adult medusa that are visible through the body wall as four horseshoe-shaped structures. Oocytes at different stages of oogenesis can be found within the same 'gonad' (*Eckelbarger & Larson, 1988*; *Hargitt & Hargitt, 1910*). The pre-mature and mature oocytes of *A. aurita* are about 120–180 µm in diameter (average diameter is $159.91 \pm 16.25$ µm ($N = 167$); the season of 2020), with the large pronucleus (germinal vesicle) located right at the animal pole (Fig. 1B). Upon maturation, while the oocyte is still in the 'gonad', the pronucleus breaks and two minute polar bodies are released (Fig. 1C).

We observed multiple oocytes (zygotes) in the brood pockets, occasionally observed the initial stages of cleavage in the gastric cavity (Fig. 1D). Within the 'gonads' only oocytes were observed. The delicate vitellin envelope becomes visible only during early cleavage (Figs. 1F and 1I). It covers the embryo until the gastrula stage, disintegrates in the course of embryonic development, and is rarely preserved during fixation procedure. The embryos developing inside the oral arm pouches (Fig. 1A) are embedded into the mucus. In natural conditions, the embryos start to move only after the reaching the planula stage. However, *in vitro*, when released from the mucus, embryos can move at the gastrula stage. Several dozens of embryos at the different developmental stages, from fertilized oocytes to early planulae, develop inside the same brood pocket (Fig. 1A).

Under controlled laboratory conditions (10−12 °C), the zygote develops to the early planula-larva within approximately 4 days, and gastrulation takes about 24 h.

We suggest the following key stages of *A. aurita* early development: cleavage, blastula, pre-gastrula, early gastrula, mid-gastrula, late gastrula, pre-planula with elongated oral-aboral axis, and planula. We investigated each of these stages with the focus on gastrulation using confocal and electron microscopy.

## Cleavage and the blastula stages

A zygote undergoes holoblastic cleavage by unilateral cleavage furrows typical for cnidarians (Figs. 1D and 1E). One cleavage cycle takes about 4 h at 10−12 °C. In the majority of embryos, the cleavage proceeds as equal and synchronous (*e.g.*, Figs. 1E, 1F, 1H–1J and 1N). However, quite often, the first and/or the second cleavages are unequal, that leads to unequal size of early blastomeres (Figs. 1G and 1K). Some embryos consist of an odd number of blastomeres, which differ in size from each other (Figs. 1L and 1O). This may be explained by the asynchrony of cleavage divisions caused by the difference in the size of blastomeres: in larger blastomeres cytotomy might proceed slower than in the smaller ones. In general, the cleavage pattern remains regular, and at the 4–16 cell stages embryo is characterized by compact cell packing with the crosses between the non-sister cells, which morphologically resembles the pattern of spiral cleavage (Figs. 1J and 1N). Embryos lost compact cell packing when they were isolated from the vitellin envelope and mucus under laboratory conditions (Figs. 1K and 1M). Nevertheless, by the 32-cells stage all embryos acquire almost spherical shape.

At the 8–16-cells stage (Figs. 1N and 2A) a small blastocoel appears (Figs. 2B and 2C). In the 16–64 cell embryos the shape of the blastomeres depends on the cell cycle stage. In interphase blastomeres are conical (Fig. 2B). During mitosis, blastomeres become almost spherical with convex apices (Figs. 2A and 2C). During the transition to the 128-cell stage, the blastocoel becomes more spacious, and the blastomeres change their shape from the conical to a more columnar; cell apices become flattened, and the entire surface of the embryo becomes more even (compare Figs. 2G, 2H, and 2I). The cilia' basal bodies become visible between the stages of 128 and 256 cells (Figs. 2I and 2J), although cilia appear on the cell apices later in development.

At the 8–16 cell stage, nuclei and mitotic spindles are located nearly in the center of the blastomeres (Figs. 2B and 2C). From the 2nd and up to the 5th round of cleavage, the unilateral furrows originate on the contact surfaces of blastomeres and spread toward the blastomere's outer surfaces (Figs. 1M and 1N). From the 32-cells stage, the nuclei are located right beneath the apical surfaces of the blastomeres, and the mitotic spindles form there too, oriented tangentially (Figs. 2D, 2F and 2K). Cleavage remains synchronous (Fig. 2F), but the cleavage furrows originate on the outer surfaces of the blastomeres and spread toward the center of the embryo (Fig. 2K). Cell divisions occur by centripetally oriented unilateral furrows until the 128–256 cell stage (Figs. 2H–2J). At the end of the blastula stage, the blastoderm cells are columnar, about 35–50 μm in height; the number of blastomeres reaches about 1,000 (Fig. 2M). In some embryos, one or several abnormally large rounded cells with the fragmented chromatin were found (Fig. 2L). Later in development, similar cells (or cell debris) were detected in the blastocoel and in the gastrocoel (Figs. 2I and 2L; 3A).

## Gastrulation

Development from a zygote to the gastrula stage takes about 36 h. Right before the onset of gastrulation, the blastula has an almost spherical shape (Fig. 2M). At the pregastrula stage, the cells on one side of the blastula elongate along their apico-basal axes concurrently

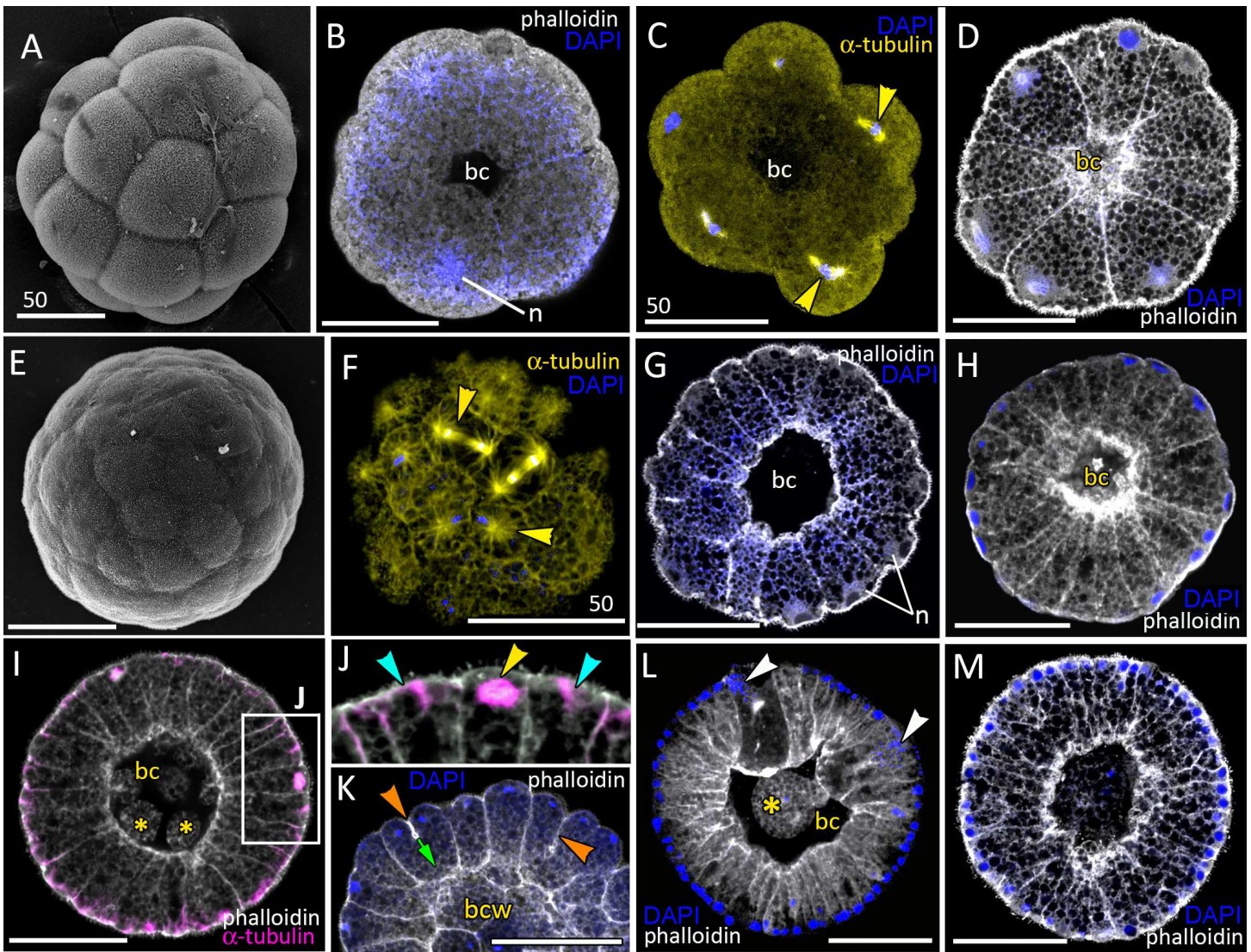

**Figure 2  Blastula stage of *Aurelia aurita*.** (A–C) 16-cell stage, beginning of blastocoel formation. (A) SEM of an embryo. (B) Embryo with conically shaped interphase blastomeres. (C) Embryo with rounded cells forming the mitotic spindles (yellow arrowheads). (D) 32-cell stage embryo with small blastocoel; cells are conical with long lateral (contact) sides. (E–G) 64-cell stage embryos; mitotic spindles (yellow arrowheads) are formed synchronously (F); blastocoel volume increases (G). (H) 128-cell stage embryo. (I, J) 256–512-cell stage embryos. (J) Subapical region of cells framed in (I); blue arrowheads point to the basal bodies of forming cilia, yellow arrowhead shows mitotic spindle. (K) 7th round of cleavage; blastula cells continue to form unilateral furrows (orange arrowheads) spreading towards the basal cell surface (green arrow). (L) Blastula with large cells containing fragmented DNA instead of nuclei (white arrowheads). (M) Embryo at the late blastula stage (more than 1,000 cells). Abbreviations: bc, blastocoel; bcw, blastocoel wall; n, nucleus. Yellow asterisks mark spherical cells (cell fragments) in blastocoel. SEM: A, E; CLSM: B–D, F–M. All scale bars: 50 μm.

with the constriction of their apical surfaces (Figs. 3A, 3C and 3D, d). That leads to a pronounced local thickening and flattening of the blastoderm (Figs. 3A, 3C and 3D, d). It is the first morphological sign of the onset of gastrulation. The blastopore will be formed in the flattened area afterward. The cells of the rest blastoderm are variable in shape, but most of them have a wedge shape with the narrow basal end (Figs. 3A and 3B). All blastoderm

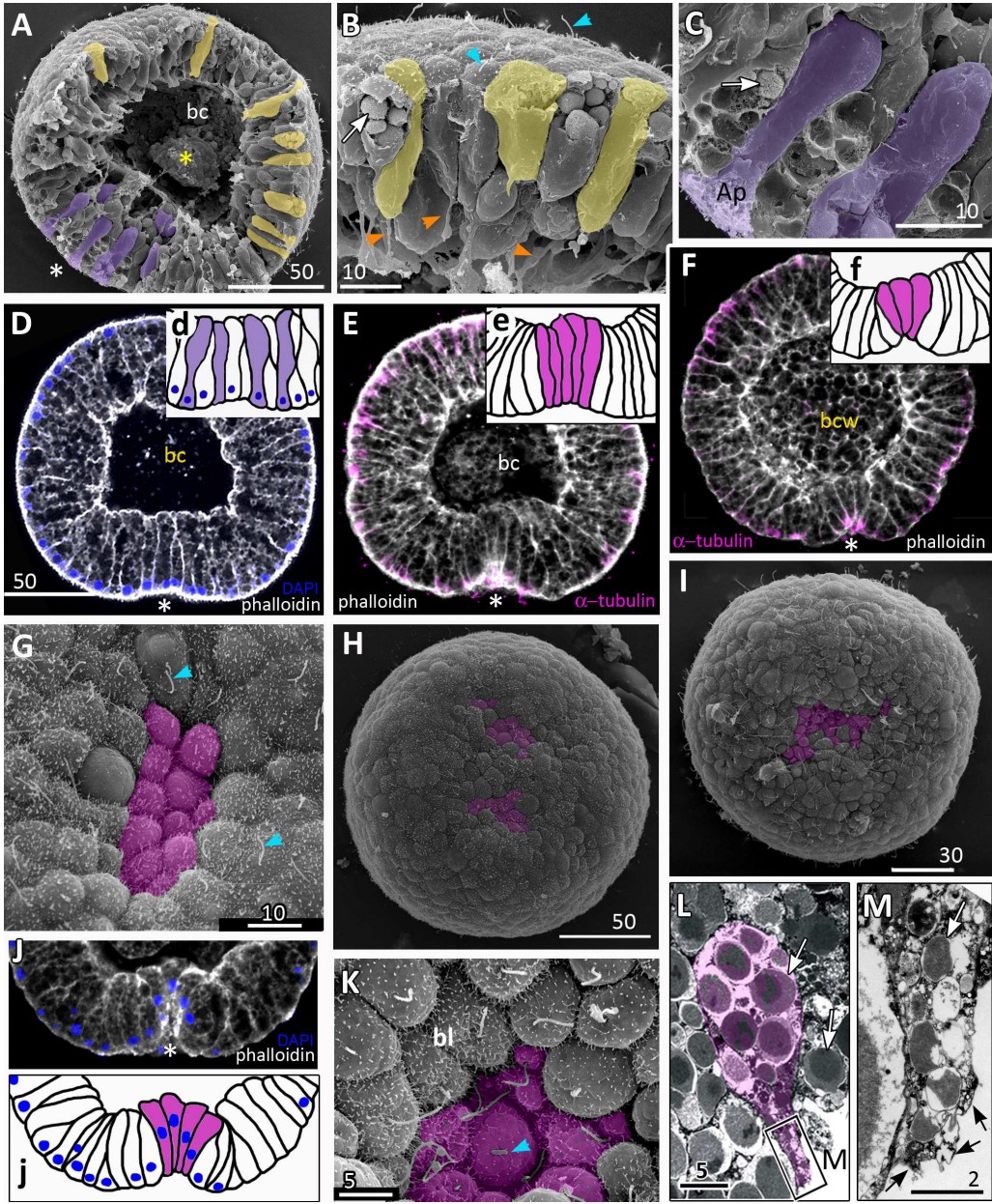

**Figure 3** *Aurelia aurita* **embryos at the pregastrula and early gastrula stages.** (A–D) Pregastrula stage. (A, D) Embryos with a flattened area composed of columnar cells. This area corresponds to the oral pole where the blastopore will form at the next stage. (A) SEM of the exposed surface of a split pregastrula. Some cells of the blastocoel roof are highlighted in yellow; cells of the flattened area are in lilac; yellow asterisk marks the rounded cell within the blastocoel. Fragments of blastocoel roof (B) and flattened area (C) are shown at higher magnification; orange arrowheads—basal outgrowths of blastocoel roof epithelial cells; white arrows—yolk granules. (D, d) A fragment of the oral area is drawn on (d); several cells with elongated apico-basal axis are highlighted in lilac. (E–K) Early gastrula, beginning of invagination. The oral areas of E, F, and J drawn correspondingly on e, f, and j; the bottle cells are highlighted in magenta. (G) Constricted apices of the bottle cells (magenta) at the bottom of the forming blastopore. (continued on next page...)

**Figure 3 (…continued)**
(H) Two domains of cells with constricted apices (magenta) are located in the flattened region of an embryo. (I) Early gastrula with an irregularly shaped domain of bottle cells (magenta). (J, j) Oral area of early gastrula. (K) Cells of the forming blastopore lip surround cells with constricted apices, which retain cilia. (L) Bottle cell filled with yolk granules (white arrows); its apical region framed and shown at higher magnification in (M); black arrows indicate the folds of the apex; white arrow points to the yolk granule. Abbreviations: Ap, the apical domain of a cell; bc, blastocoel; bl, blastopore lip; bcw, blastocoel wall. The white asterisk marks the area of blastopore formation; blue arrowheads indicate cilia. SEM: A–C, G–I, K; CLSM: D–F, J; TEM: L, M.

cells form intertwining basal processes (orange arrowheads in Fig. 3B) and are filled with the large yolk granules (white arrows in Figs. 3B and 3C). The cilia appear at the pregastrula stage (blue arrowheads in Fig. 3B).

Further on, the cells situated in the centre of the flattened region change their shapes from the columnar to the bottle-like (Fig. 3E, e). In the bottle cells, contraction of apices is accompanied by the widening of the basal domains and shortening of the apico-basal axes (Figs. 3F, f and 3J, j). Apico-basal axes of bottle cells become even shorter than in the neighbouring blastoderm cells (Fig. 3F, f). The change in cell shape results in appearance of a depression in the centre of the flattened area, indicating that the early gastrula stage has been reached (Figs. 3E, 3F, 3G and 3J, j).

On the surface of the pregastrula stage embryos the groups of cells with constricted apices are easily distinguishable (Figs. 3G, 3H and 3I). Many embryos have several (from 2 to 4) groups of such cells (Fig. 3H). Apparently, during further development these groups merge together as the cells in between the groups acquire a bottle shape too. In the early gastrulae, 10–25 cells with constricted apices are organized into one group (average number of cells is $17.5 \pm 4.5$; $N = 10$ embryos) (Figs. 3G and 3I; 4A and 4B).

At the beginning of gastrulation, the archenteron on a section has a shape of a low arc (Figs. 3F and 3J; 4C). The bottle cells with constricted apices and widened basal domains occupy the top of the archenteron (Figs. 3F and 3J, j; 4C). Nuclei in these cells are shifted into the basal domains (Fig. 3J, j). The cells constituting the developing archenteron contact with each other, but not with the inner surface of the blastocoel wall (future ectoderm); they have no pseudopodia on their basal surfaces. Bottle cells retain cilia (Fig. 3K). The cytoplasmic membrane of the apical surfaces of bottle cells forms numerous folds (Figs. 3L and 3M). Blastoderm cells surrounding the bottle cells represent the forming blastopore lip (Fig. 3K).

At the mid-gastrula stage invagination continues, and the deep blastopore surrounded with the blastopore lip forms (Figs. 4A–4F, 4H, 4K and 4M). Cells of the blastopore lip differ markedly from the cells of the archenteron. These cells have a very pronounced wedge shape, and their nuclei locate near the cell apical surfaces (Figs. 4E, 4F and 4H, h). The apical surfaces of these cells are elongated towards the blastopore (Fig. 4C, c). The apicobasal axes of these cells bend towards the blastopore that becomes apparent at the early gastrula stage (Figs. 4C, c, 4E, e, 4G and 4H, h; blue cells in Fig. 4K). The shape and behaviour of the archenteron cells are very diverse at these stages. The cells located at the top of the archenteron often acquire a rounded or a cuboidal shape due to the contraction along their apicobasal axes (brown cells in Figs. 4H–4J). The cells situated at the base of

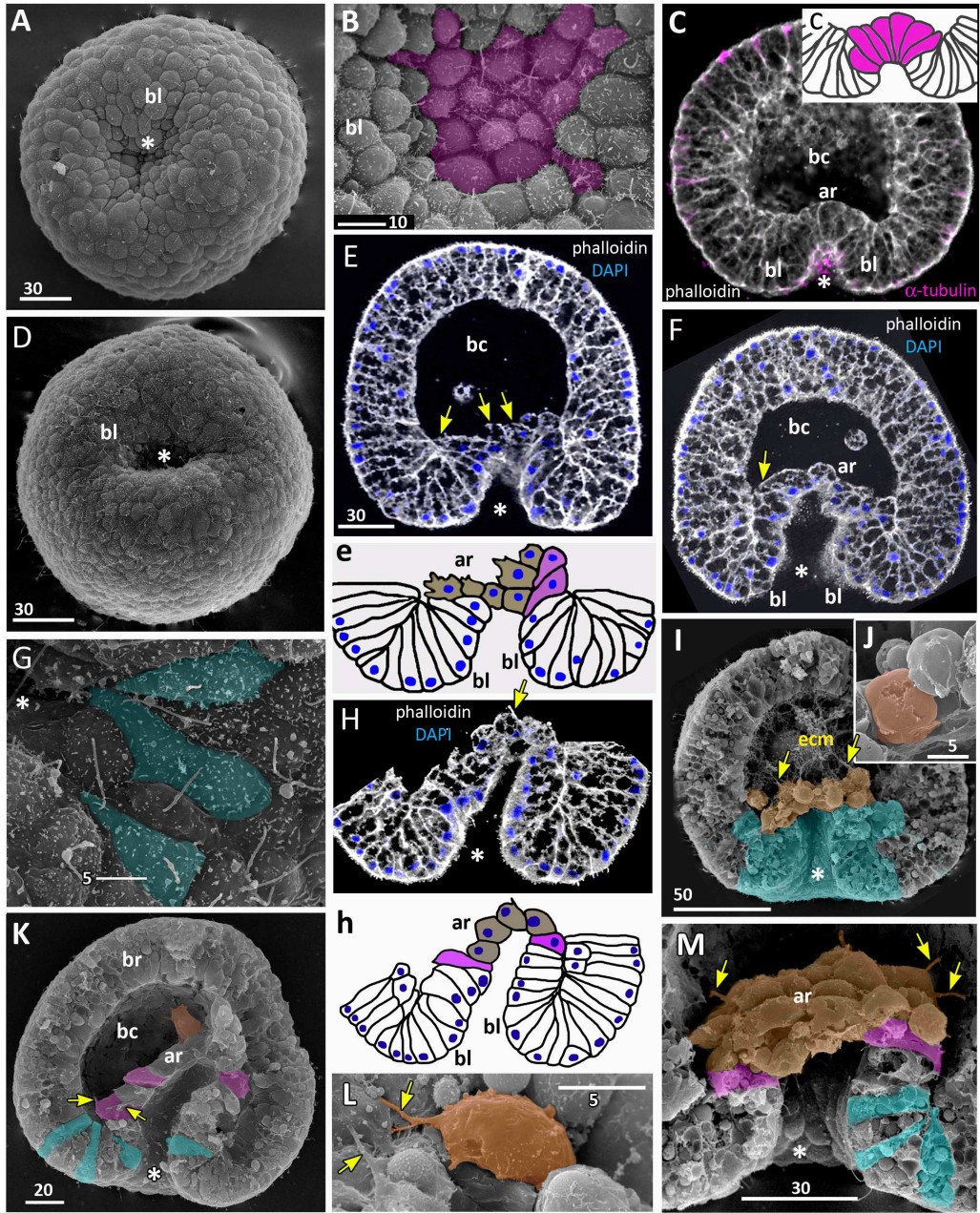

**Figure 4** *Aurelia aurita* **embryos at the mid-gastrula stage.** (A–F) Gradual deepening of the blastopore, formation of archenteron and blastopore lip. (A) Embryo with a shallow blastopore, the tip of the invaginating archenteron is still visible from the outside. (B) The tip of the archenteron consists of about 20 cells with constricted apices. (C, E, F, H) Optical sections of embryos at successive stages of archenteron invagination. (C) Optical section of an embryo at the stage shown in (A). The blastopore areas of the embryos shown in (C, E) drawn correspondingly in (c, e). (D) Embryo with a deep blastopore; the tip of the archenteron is not visible from the outside; the blastopore lip is steep. (continued on next page...)

**Figure 4 (...continued)**
(G) The apices of the blastopore lip cells are elongated towards the blastopore; three apices are highlighted in blue. (H-M) Images showing the shape of archenteron and blastopore lip cells in mid-gastrula stage embryos. (H) Optical section through the blastopore area; this section schematically drawn in (h). (I) SEM of the exposed surface of a split embryo; all archenteron cells have lost their bottle shape; archenteron cells extend filopodia toward extracellular matrix. (J) Rounded cells at the archenteron tip. (K) SEM of the exposed surface of a split embryo; the majority of archenteron cells retain the columnar or bottle shape. (L, M) Leading edges of archenteron cells, view from the blastocoel. (L) Leading edges of cells, which occupy the archenteron tip; one of the cells is highlighted in brown. (M) Archenteron tip, leading edges of archenteron cells extend filopodia toward blastocoel wall. Abbreviations: ar, archenteron; bc, blastocoel; bl, blastopore lip; ecm, extracellular matrix. The white asterisk marks the blastopore area. Archenteron cells that have retained the bottle shape are highlighted in magenta, archenteron cells that have lost the bottle shape are highlighted in brown; the blastopore lip cells are highlighted in blue; yellow arrows indicate leading edges of cells (E, F, H) or filopodia (I–M). SEM: A, B, D, G, I–M; CLSM: E, F, H.

the archenteron retain the bottle shape (magenta cells in Figs. 4E, e, 4H and 4M). In some embryos, the archenteron consists of columnar cells with short apicobasal axes and bottle cells scattered between them (Fig. 4K).

At the mid-gastrula stage, the archenteron cells form multiple protrusions on their leading edges (yellow arrows in Figs. 4H, 4I, 4L, and 4M). The cells extend filopodia and lamellae toward the blastocoel wall (Fig. 4M), to the extracellular matrix filling the blastocoel (Figs. 4H and 4I), and to each other (Fig. 4L).

At the beginning of the late gastrula stage (Fig. 5A), the archenteron occupies about 2/3 of the blastocoel space (Fig. 5B). When gastrulation is almost accomplished, the archenteron cells get in contact with the blastocoel roof, and the rest of the blastocoel disappears (Figs. 6B–6E).

Archenteron cells of late gastrulae demonstrate very pronounced migratory behaviour. Such cells form the distinct leading edge with multiple protrusions (Figs. 5D and 5G). They crawl aborally along the blastocoel wall maintaining contact with neighbouring archenteron cells (Figs. 5E, 5F and 5H).

At the late gastrula stage (Fig. 5), the morphology of embryos is very diverse. In some embryos, archenteron does not represent a continuous structure. Figures 5E and 5H show an embryo with large spaces between the archenteron cells migrating along the blastocoel wall. Archenteron cells contact their neighbours through the multiple filopodia-like protrusions formed at their leading and trailing edges (Fig. 5G, magenta arrows show the trailing edge protrusions). The most aboral archenteron cells acquire a flattened shape characteristic of a mesenchyme cell. They constitute the migratory front that pulls the rest of the archenteron towards the aboral pole (white arrows in Fig. 5E).

In the embryo shown in Figs. 5B and 5F, the dome-shaped archenteron did not lose its integrity. In this embryo, the archenteron cells cover almost an entire surface of the blastocoel roof. The archenteron is composed of cuboidal cells, and several cells situating near the blastopore lip retain the bottle shape (magenta cells in Figs. 5C and 5H).

The presumptive ectoderm (blastocoel roof) gains a structure of a pseudostratified epithelium. Basally located nuclei appear in the blastoderm apart from the blastopore region from the early gastrula stage (black arrows in Figs. 5B; Figs. 6C and 6E). The number of such nuclei increases, and, as it becomes evident at the late gastrula stage, they

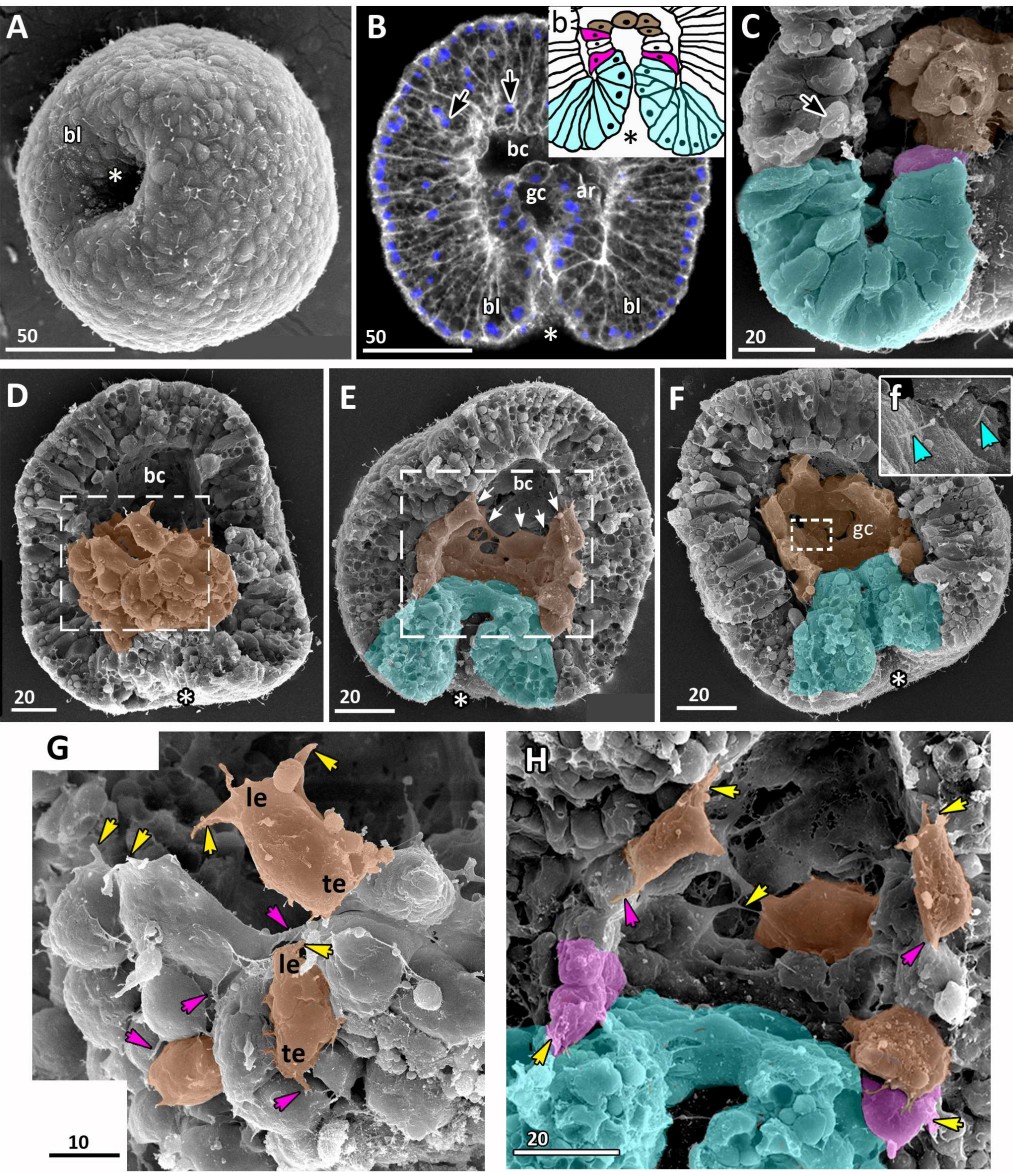

**Figure 5  Late gastrula stage of _Aurelia aurita_ embryos: archenteron moves towards the aboral pole.**
(A) SEM of an embryo with the wide opening of the blastopore. (B) Optical section through an embryo;
the tip of the archenteron does not come into contact with the blastocoel roof, and the blastocoel is
still present; black arrows show the nuclei locating at the base of the ectoderm (DAPI staining—blue,
phalloidin—white). (b) A schematized drawing of the blastopore area of an embryo presented in (B);
cuboidal/rounded cells at the tip of the archenteron are brown, wedge-shaped archenteron cells are
white, bottle cells are magenta, blastopore lip is blue. (C) Fragment of exposed surface of embryo split
into halves showing blastopore lip (highlighted in blue) and part of archenteron (magenta and brown
cells); black arrow indicates a small rounded cell at the base of the ectoderm. (D–H) Crawling behaviour
of the archenteron cells. (D) SEM of a late gastrula with ectoderm partially removed. The surface of
the archenteron exposed to the blastocoel is visible; the archenteron cells (presumptive endoderm)
highlighted in brown. (continued on next page...)

**Figure 5 (…continued)**
(E, F) Variability of archenteron structure; SEM of exposed surfaces of embryo split into halves; blastopore lip is highlighted in blue. (E) The archenteron is discontinuous, with most aboral cells forming the migratory front (white arrows). (F) Embryo with a continuous archenteron covering almost the entire basal surface of the blastocoel roof. (f) Higher magnification of the fragment framed on (F); blue arrowheads point to the cilia of the archenteron cells. (G) Archenteron cells, embryo fragment framed on (D); yellow arrowheads indicate protrusions on leading edges of crawling archenteron cells, magenta arrowheads— protrusions on trailing edges of these cells. (H) Archenteron cells crawling over blastocoel roof, embryo fragment framed on (E); crawling cells with amoeboid morphology are highlighted in brown, bottle cells are magenta. Abbreviations: ar, archenteron; bc, blastocoel; bl, blastopore lip; gc, future gastrocoel; le, cell leading edge; te, cell trailing edge. The asterisk marks the blastopore. SEM: A, C, D–H; CLSM: B.

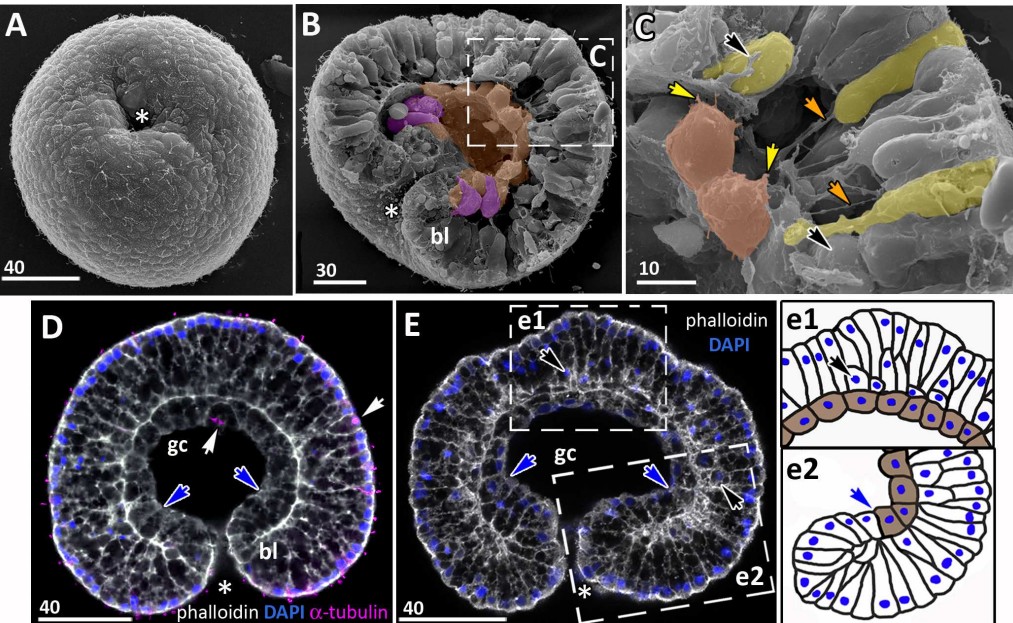

**Figure 6** **Late gastrula stage of *Aurelia aurita* embryos: the archenteron gets in contact with the blastocoel roof.** (A) Late gastrula with the narrow blastopore opening. (B) SEM of the exposed surface of an embryo split into halves. The archenteron is in close contact with the blastocoel roof; most of the archenteron cells are cuboidal or rounded (highlighted in brown); the cells near the blastopore lip retain the bottle shape (magenta). (C) Higher magnification of the fragment framed on (B); some ectodermal cells are colored in yellow, endodermal cells are brown; yellow arrowheads indicate protrusions on former leading edges of archenteron cells, orange arrowheads point to basal protrusions of ectodermal cells, black arrowheads indicate small rounded cells locating at the base of the ectoderm. (D, E) Central optical sections of the late gastrulae; dark-blue arrowheads indicate the boundary between the blastopore lip and the archenteron. (D) Embryo with the archenteron composed of relatively large cuboidal cells; white arrowheads point to the mitotic spindles (Image credit: Stanislav Kremnyov). (E) Embryo at a slightly later stage than in (D), the number of endodermal cells has increased; black arrowheads indicate the basally located nuclei in the ectoderm; framed fragments of the embryo are schematically drawn and shown in (e1) and (e2); the endoderm in (e1) and (e2) is brown. Abbreviations: bl, blastopore lip; gc, future gastrocoel. The asterisk marks the blastopore. SEM, A–C; CLSM, D, E.

belong to small rounded cells occupying the space in between intertwining basal processes of the blastoderm cells (Figs. 5C and 6C).

At the late gastrula stage, the number of the endodermal cells increases (Fig. 6D). By the end of gastrulation, the endoderm consists of about 40–50 cuboidal cells adjoining the basal surface of the ectoderm (Figs. 6B–6E). The blastopore opening gradually becomes smaller (compare Figs. 5A, 5F, and 6A). Finally, the opposite sides of the blastopore lip get in contact with each other. The closed blastopore remains visible from the oral surface (Figs. 7A and 7G) and in sections (Figs. 7E, 7F, and 7H). The embryo reaches the preplanula stage.

### The preplanula stage, development of the planula-larva

During the preplanula stage, features of the planula-larva gradually appear. Development of the planula includes the following processes: elongation of the oral-aboral axis; "healing" of the closed blastopore; increasing in the number of the endodermal cells; cell differentiation.

Gradual elongation of the oral-aboral axis is shown on the SEM images of embryos at the stages of late gastrula (Fig. 6A) and preplanula (Figs. 7A and 7G). Elongation of the oral-aboral axis is accompanied by the morphological differentiation of the axis poles: the oral pole becomes pointed while the aboral pole keeps a rounded shape (Figs. 7J–7K). A preplanula is capable of active changes of its body shape (compare preplanula shape in Figs. 7J and 7K).

"Healing" of the closed blastopore proceeds gradually (Figs. 7C–7J). Right at the moment of the blastopore closure, the epithelium of the blastopore lip seamlessly passes into the presumptive endoderm (Figs. 6E, e2; 7C and 7L). At the next stage, the border between the cells of the blastopore lip and the cells of the presumptive endoderm becomes more pronounced (Figs. 7D, 7E and 7M). Then the endoderm becomes a continuous layer (Figs. 7F, 7H and 7N). At the final steps of the blastopore healing, the scar from the closed blastopore disappears, and the ectoderm becomes a continuous layer too (Figs. 7I, 7J, and 7K). Integrated into the ectoderm, cells of the blastopore lip occupy the oral pole of the embryo.

During preplanula development, endodermal cells change their shape from the cuboidal to a more columnar (compare the cells in Figs. 6C, 6D, 6E, e1, 7C, 7D, 7J and 7K) with nuclei located in the apical domain (Fig. 7D). The endodermal epithelium remains monostratal. At the end of the preplanula stage, the embryo acquires a tear-like shape with the broader and more rounded aboral (anterior) pole (Figs. 7I–7K), the length/width ratio less than 2, and a pronounced gastric cavity. Separation of germ layers and morphological differentiation of oral and aboral poles can be considered the final steps of preplanula development. In the laboratory, the embryo starts swimming at the preplanula stage.

### Some features of the planula structures

When the preplanula transforms into the planula, it elongates further and acquires a sausage-like shape (Figs. 8A and 8E). The planula moves with its aboral pole forward, and this pole is characterized by a slightly flattened shape and an apical tuft of elongated cilia (Figs. 8B–8D). Planula larva that is ready for settlement and metamorphosis (*i.e.*, competent planula) leaves the oral arms of medusa.

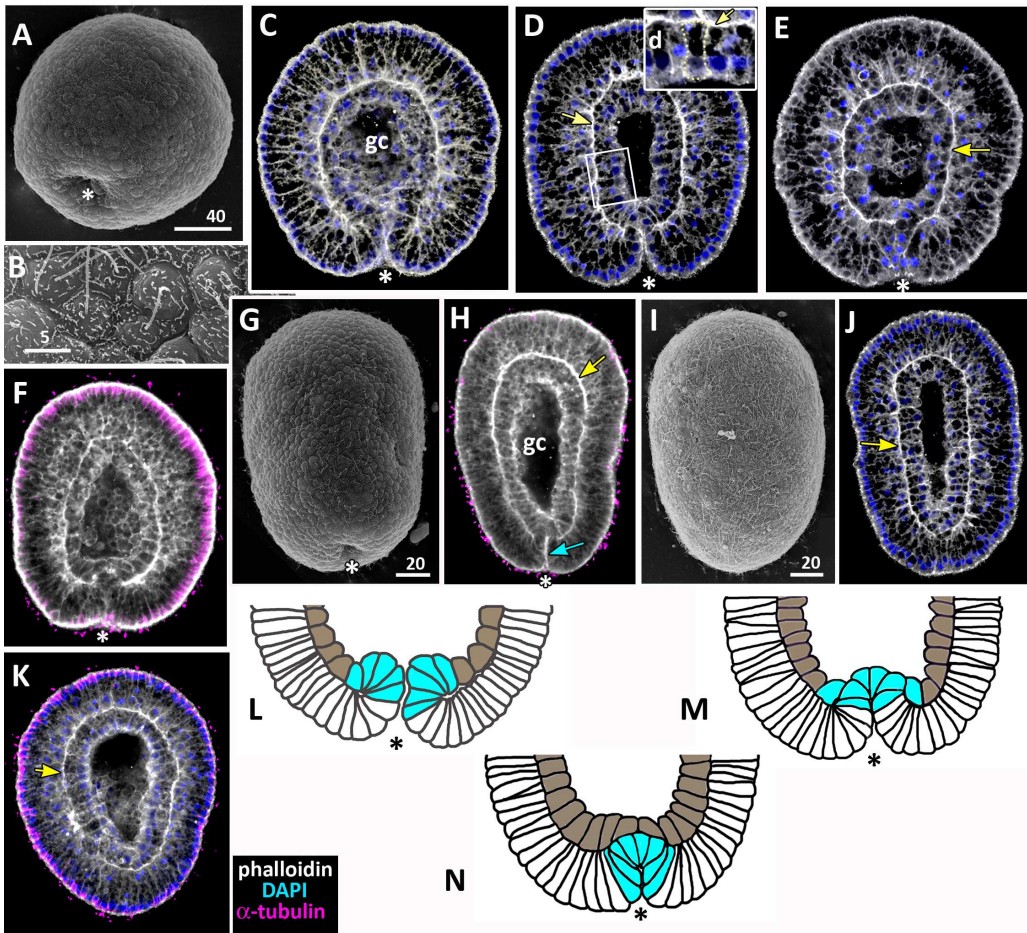

**Figure 7** **Developing preplanula of *Aurelia aurita*.** (A) SEM of an embryo with a nearly closed blastopore. (B) SEM of the surface of the embryo shown in (A). (C–K) Oral pole is down. (C) Optical section through the embryo similar to the embryo shown in (A), the oral-aboral axis of the embryo is slightly elongated. (D-J) Successive stages of blastopore closure. First, a continuous endoderm forms (compare D, E, F, H) and then continuous ectoderm (compare D, H and J); yellow arrows show the basal lamina. (d) Endodermal columnar cells framed on (D) at higher magnification; the shape of one of the cells is accentuated with the yellow dotted line. (G, H) The opening of the blastopore remains visible for a long time; blue arrow points to the scar from the closed blastopore opening, the ectoderm has not yet closed here. Simultaneously with the closure of the blastopore, the oral-aboral axis elongates (compare C, D and H, J). (K) Preplanula with pointed oral (posterior) end. (L–N) Scheme of the successive stages of blastopore closure; the blastopore lip is shaded blue, the archenteron cells are brown. Scheme (L) corresponds to the embryo shown in (C), scheme (M)—to the embryos in (D, E) and scheme (N)—to the embryos in (F, H). Abbreviations: gc, future gastrocoel. The asterisk marks the closed blastopore. SEM: A, B, G, I; CLSM: C–F, H, J, K. DAPI—blue, phalloidin—white.

The length of the competent planula ranges from 200 to 350 μm, the width—from 110 to 150 μm, and the length/width ratio—from 2 to 3.5. The exact sizes vary in different samples. In 2020, the competent planulae obtained from two females had an average length of 233.31 ± 20.46 μm and an average width of 126.83 ± 7.09 μm ($N = 115$).

Here we give a brief description of the tissue and cellular organisation of *A. aurita* planula. The planula is built of two epithelial layers composed of several cell types, has

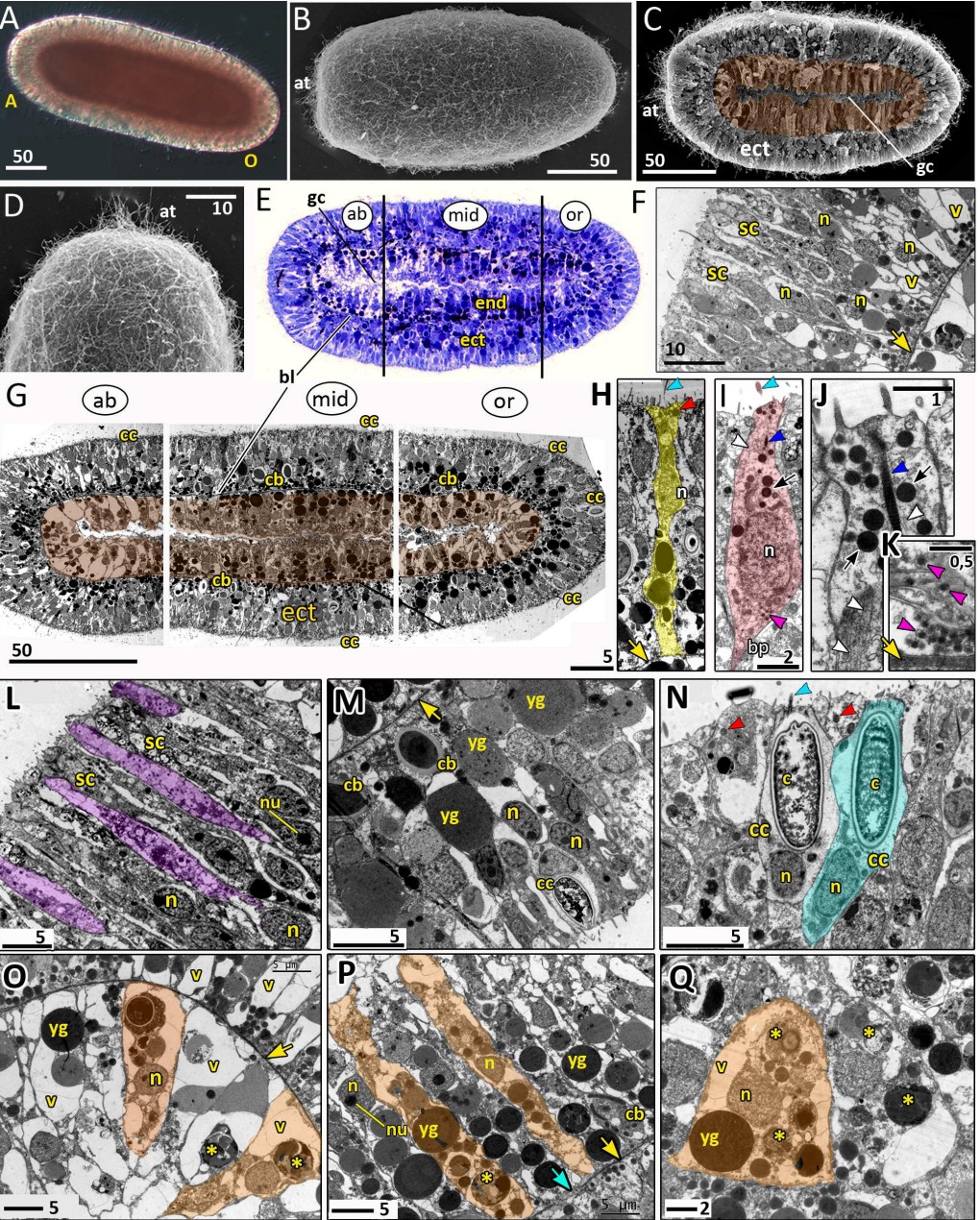

**Figure 8** **Planula larva of *Aurelia aurita*.** The oral pole is on the right at (A, B, C, E, G). (A) Light microscopy image of a planula. (B) SEM of a planula. (C) Planula split into halves; endoderm is artificially colored in brown. (D) Apical tuft area at higher magnification showing elongated cilia. (E) Semithin longitudinal section of a planula; black lines divide the planula body into three parts according to differences in the endoderm structure. (F–N) TEM of planulae. (F) Aboral ectoderm of a planula. (G) TEM of the three regions of the planula shown in (E). (H) Epithelio-muscular (supportive) cell of the ectoderm (highlighted in yellow). (I) Putative nerve cell of the ectoderm (highlighted in pink). (J) Apical domain of a putative nerve cell. (K) Basal area and basal processes of a putative nerve cell filled with the characteristic dense-core vesicles. 

**Figure 8 (...continued)**
(L) Aboral ectoderm cells at high magnification, vacuoles of secretory cells are artificially colored in violet. (M) Lateral ectoderm cells at high magnification. (N) Oral ectoderm cells; cnidocyte is highlighted in blue. (O) Aboral endoderm cells; two cells are artificially colored in orange. (P) Lateral endoderm cells; blue arrow points to break in basal lamina. (Q) Aboral endoderm cells; one of the cells is highlighted in orange. Abbreviations: A, aboral pole; ab, aboral part of a planula; at, apical tuft; bl, basal lamina; bp, basal process; c, cnidocyst; cb, cnidoblast; cc, cnidocyte; ect, ectoderm; end, endoderm; gc, gastrocoel; mid, middle part of a planula; n, nucleus; nu, nucleolus; O, oral pole; or, oral part of a planula; sc, secretory cells; v, vacuoles; yg, yolk granule. Yellow asterisks mark phagosomes; yellow arrowheads indicate basal lamina; blue arrowheads show cilia; dark blue arrowhead—cilium' rootlet; white arrowhead—bundles of microtubes; red arrowheads point to the electron-dense vesicles in the cell apical domain; magenta arrowhead—dense-core vesicles; black arrow—electron dense vesicles in putative nerve cells. Light microscopy: A; SEM: B-D; semi-thin section: E; TEM: F-N.

well-developed basal lamina separating these layers, and a pronounced gastric cavity (Figs. 8C, 8E–8Q).

The majority of the ectodermal cells are the epithelio-muscular (supportive) cells, which are typical for cnidarians. These are slender flagellated cells containing electron-clear and electron-dense vesicles within the apical cytoplasm (Figs. 8G, 8H and 8N). There are numerous yolk granules and large transparent vacuoles in the basal part of these cells (Figs. 8F, 8G, 8H and 8M). In the majority of cells, the nucleus locate in the apical domain (Fig. 8H).

In the ectoderm, spindle-shaped cells are found between the supportive cells (Figs. 8I–8K). In the apical cytoplasm, these cells contain bundles of microtubules and numerous electron-dense spherical vesicles (Figs. 8I and 8J). A thin process extends from the body of the cell towards the basal lamina (Fig. 8I). The basal part of the cell body and the basal cell process contain multiple dense-cored vesicles (Figs. 8I and 8K). The morphology and ultrastructure of these cells allow considering them as nerve (neurosecretory, sensory) cells (*Lesh-Laurie & Suchy, 1991*).

The aboral (anterior) ectoderm of the planula is characterized by a high number of gland (secretory) cells with large inclusions in their apical parts (Figs. 8E–8G and 8L). The inclusions of the most common type consist of densely packed vesicles with fibrous contents, often with an electron-dense spot in its center (Fig. 8L). The nuclei in the aboralmost cells locate in the middle or basal part of the cell (Figs. 8F and 8L). Cnidocytes are located predominantly in the lateral and posterior ectoderm of the planula (Figs. 8G, 8M and 8N). The numerous clusters of cnidoblasts at the different stages of cyst development are scattered in the basal area of the lateral ectoderm (Figs. 8G and 8M). The oral ectoderm contains very few cnidoblasts. The aboral ectoderm contains cnidocytes except for the apical most region, which is free from both cnidocytes and cnidoblasts (Figs. 8G and 8F).

The endodermal cells of the planula are filled with numerous yolk granules and phagosomes (Figs. 8G, 8O–8Q). Nuclei locate in the middle part of the endodermal cells. They usually have non-condensed chromatin and nucleoli (Fig. 8P). The planula body can be subdivided into three well-defined compartments along the oral-aboral axis according to the morphology and ultrastructure of the endodermal cells (Figs. 8E and 8G).

The aboral (anterior) compartment (Figs. 8E, 8G, and 8O) is characterized by densely packed cells with highly vacuolated cytoplasm. The nucleus is located in the center of the

cell. The cells have a conical shape at the aboral pole (Fig. 8O) and a columnar shape on the sides of the planula body (Fig. 8G). The basal lamina in the aboral compartment is up to 0.2 μm thick,

The middle compartment (Figs. 8E, 8G, and 8P) constitutes up to half of the planula endoderm. The cells are columnar and narrow with small vacuoles (Figs. 8G and 8P). The basal part of the cytoplasm is filled with yolk granules of different sizes. The nucleus is located in the middle part of the cell (Fig. 8P). There are numerous cnidoblasts at the different stages of cyst development, from the Golgi vesicles to completely developed capsules with the thread (Fig. 8P). Usually, the developing cysts locate in the typical cnidoblasts, but sometimes they can be found inside the epithelio-muscular cells. The basal lamina within the middle compartment is almost continuous, with only occasional gaps (Fig. 8P).

Endodermal cells of the oral (posterior) compartment are vacuolated but less than the oral cells (Figs. 8E, 8G, and 8Q). Yolk granules occupy the basal part of the cell, numerous phagosomes are evenly distributed in the cytoplasm. The nucleus is located predominantly in the middle part of the cell. Cnidoblasts are present in the oral endoderm, but they are not numerous. The oral basal lamina is approximately 2–3 times thinner than in the aboral part of the planula (Fig. 8Q).

## DISCUSSION

In this work, we have described successive stages of embryonic development of *A. aurita*. We summarized the key stages and events as follows: cleavage and early blastula (Fig. 9A); late blastula (Fig. 9B); pregastrula (Fig. 9C); early gastrula (beginning of archenteron invagination) (Figs. 9D and 9E); mid-gastrula (Fig. 9F); late gastrula whose morphology is very variable (Figs. 9G1–9G3); preplanula formation (blastopore closure, elongation of oral-aboral axis) (Figs. 9H and 9I); preplanula with closed blastopore (Fig. 9J); late preplanula and competent planula with pointed oral end and rounded aboral end (Fig. 9K).

***Setting up for gastrulation: formation of a spherical blastula and pregastrula***
Early blastula with the small blastocoel already forms at the 16–32 cell stage (Figs. 2A–2H; 9A). Acquiring the regular blastula morphology early in development is a result of the compact packing of blastomeres, provided in hydrozoans and anthozoans only by active movements of blastomeres (*Burmistrova et al., 2018*; *Conklin, 1908*; *Fritzenwanker et al., 2007*; *Kraus et al., 2014*; *Teissier, 1929*). In *A. aurita*, eggs and early embryos are covered with the vitelline envelope that can contribute to maintaining the regular cleavage and compact cell packing (Figs. 1C, 1E, 1F and 1I). This envelope has been observed in different scyphozoan species (*Claus, 1883*; *Hargitt & Hargitt, 1910*; *Metschnikoff, 1886*). According to Ikeda and colleagues, the acellular coat surrounding the developing oocyte is a portion of the basal lamina of the gastrodermis that gives rise to scyphozoan germ cells (*Ikeda, Ohtsu & Uye, 2011*). We removed mucus and accidentally destroyed the vitelline envelope when extracted the eggs and embryos from the oral arm pockets. Then we observed asymmetry

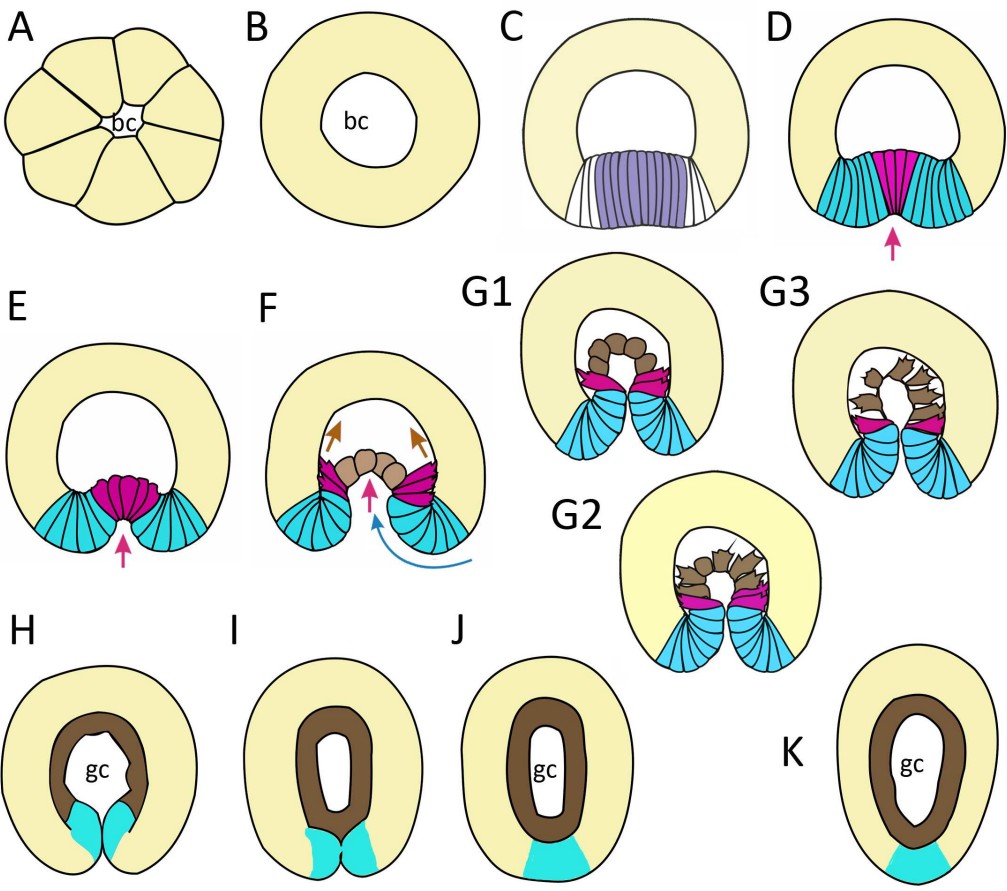

**Figure 9** **Schematic representation of the normal development of *Aurelia aurita*.** The oral pole is down at (C–K). (A) Early blastula. (B) Late blastula. (C) Pregastrula; columnar cells in flattened region are colored in lilac (cells with constricted apices) and white (future blastopore lip cells). (D, E) Early gastrula, beginning of archenteron invagination; archenteron cells have assumed a bottle shape (magenta); blastopore lip cells are blue. (F) Mid-gastrula; cells of the archenteron that have lost their bottle shape are brown; cells near the blastopore lip that have retained their bottle shape are magenta. (G1–G3) Variability in the morphology of the late gastrula. (H, I, J) Successive stages of preplanula formation (blastopore closure, elongation of oral-aboral axis); endoderm (former archenteron) is brown. (J) Preplanula with closed blastopore. (K) Late preplanula and planula with pointed oral pole and rounded aboral pole. Red arrows—invagination; blue arrows—involution of the blastopore lip; brown arrows—migration of the archenteron cells towards the aboral pole. Abbreviations: bc, blastocoel; gc, gastrocoel. In (H–K) the region colored in blue corresponds to the former blastopore lip.

and asynchrony of cleavage as well as loosely packed blastomeres more often than in the brood pockets (Figs. 1K and 1M).

Late blastula with relatively spacious blastocoel consists of 800–1,000 wedge-shaped cells (Figs. 2M; 9B; 10A, a). By this stage, cell divisions become asynchronous (Figs. 2I and 2J), and the unilateral furrows are replaced by the ordinary circular furrows. According to our data and data of other authors (*Yuan et al., 2008*), gastrulation starts when embryo consists of about 1,000 cells. During the transition from the late blastula to the pregastrula

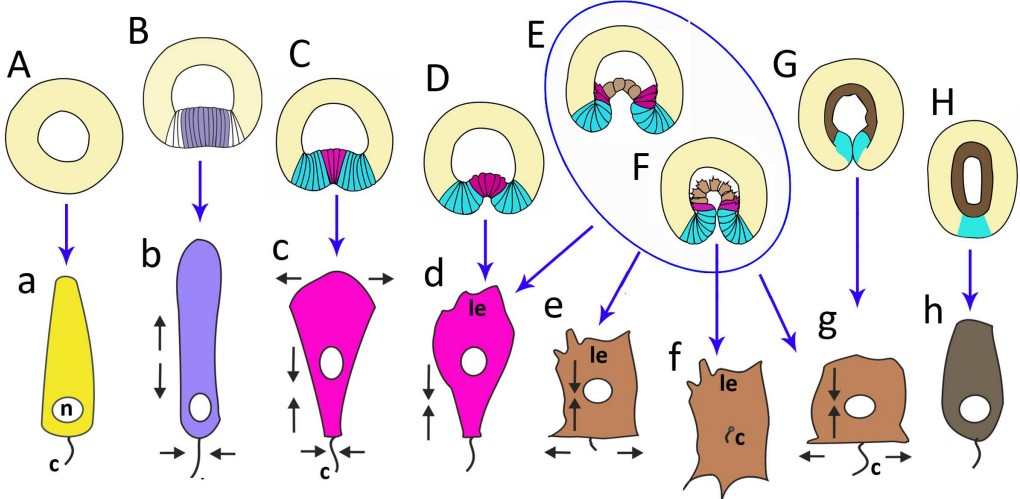

**Figure 10 The schematic shows the cell shapes in *Aurelia aurita* embryos at successive stages of development.** Embryonic stages: (A) late blastula, (B) pregastrula, (C, D) early gastrula, (E) mid-gastrula, (F) late gastrula, (G) developing preplanula during blastopore closure, (H) preplanula with closed blastopore. Color code is the same as in Fig. 9. Cells: (a) wedge-shaped cell of a blastula; (b) columnar cell of the oral region of the pregastrula; (c, d) bottle cells of the gastrula; (e) archenteron cell of the midgastrula/late gastrula with multiple protrusions on the leading edge; (f) archenteron cell with the phenotype of a mesenchyme cell and the pronounced migratory behavior; (g) archenteron cell that has assumed a cuboidal/rounded shape; (h) cell of the epithelial endoderm of the preplanula. Black arrows show the shape changes of the different regions of a cell. Abbreviations: n, nucleus; c, cilium; le, leading edge.

stage, embryo' symmetry breaks by formation of the oral domain of elongated columnar cells (Figs. 9C, 10B, b).

### Cellular mechanisms of gastrulation by invagination specific to A. aurita

Bending inward of a continuous epithelial sheet is one of the fundamental morphogenesis providing the establishment of metazoan' body plans. This morphogenesis is commonly called "tissue invagination". Tissue invagination during gastrulation has been characterized at molecular-, cellular- and mechanical levels for several model invertebrates including sea urchins (*Davidson et al., 1995*; *Ettensohn, 2020*), fruit flies (*Gheisari, Aakhte & Müller, 2020*; *Rauzi et al., 2013*; *Sweeton et al., 1991*), ascidians (*Fiuza & Lemaire, 2021*; *Sherrard et al., 2010*). It was shown that tissue invagination is based on a series of coordinated cell shape changes (*Hardin & Keller, 1988*; *Keller, Davidson & Shook, 2003*; *Leptin & Grunewald, 1990*; *Sawyer et al., 2010*; *Sherrard et al., 2010*).

According to most authors, invagination is the only mode of gastrulation in all scyphozoans (including *Aurelia*) (*Hargitt & Hargitt, 1910*; *Yuan et al., 2008*; *Smith, 1891*), and we have confirmed this conclusion. We reconstructed the changes in cell shape and cellular behavior associated with successive stages of gastrulation by invagination in *A. aurita*. These changes transform a spherical blastula into an elongated planula larva with two epithelial germ layers, gastric cavity and morphologically polarized body axis.
### Primary and secondary invagination in A. aurita

The first step of gastrulation by invagination is the 'primary invagination' (*Davidson et al., 1995*): the bending of tissue that is linked with cell apical constriction in almost all models studied in this respect (*Sawyer et al., 2010*). Shrinkage of the cell's apical perimeters bases on the activity of an actomyosin network, which locates in the cell's apical domains and produces active force for tissue flattening and bending (*Martin, Kaschube & Wieschaus, 2009*).

In *A. aurita*, primary invagination starts at the pregastrula stage with the formation of the oral domain, where wedge-shaped cells with the narrow basal ends (Fig. 10A) acquire a columnar morphology (lilac cells in Figs. 9C; 10B). Apical constriction of the cells situated in the centre of the oral domain is linked with cell lengthening along the apicobasal axis (Fig. 10B). We assume that the cell reshaping might cause flattening of the oral domain.

In the next step (at the early gastrula stage), several cells located in the centre of the oral domain acquire a bottle shape (lilac cells in Figs. 3E, j; 9D; 10C, c). These cells constrict their apical surfaces, slightly shorten the apicobasal axes, and widen the basal ends (arrows in Fig. 10C). The cell reshaping leads to the formation of a shallow depression (Fig. 3G). Then, the bottle cells further shorten along their apicobasal axis with no further apical constriction while their basal ends further expand (Figs. 3F; 4c; 9E; 10D, d). It seems that the bottle cells reduce their intercellular contacts, and, as a result, their basal ends round up and the bottle cells form a fan (Figs. 3F; 4C; 9E). That causes the depression to deepen (Figs. 4A–4D). We assume that not all the cells located in the pregastrula oral domain (lilac cells in Fig. 9C) become the archenteron cells. Many of these cells might join the presumptive blastopore lip (white cells in Fig. 9C) at the next developmental stage.

To summarize, flattening and bending of the oral epithelium are caused by the following changes in the shape of the oral domain cells: (1) columnarization and initial apical constriction; (2) coordinated apical constriction and apicobasal shortening; (3) further shortening along the apicobasal axis with no further apical constriction, and rounding up of the basal ends (Figs. 10B–10D).

Cellular mechanisms that seem to account for primary invagination in *A. aurita* were observed in other model objects (*Davidson et al., 1995*; *Keller, Davidson & Shook, 2003*). For example, two-phase invagination with apical constriction and columnarization followed by apicobasal shortening drive the ascidian endoderm invagination (*Sherrard et al., 2010*) and mesoderm invagination in *Drosophila* (*Sawyer et al., 2010*).

During secondary invagination, the archenteron moves deeper into the blastocoel and, finally, it gets into contact with the blastocoel roof. An example of secondary invagination is the elongation of the sea urchin archenteron. In sea urchin, this process is driven by traction of secondary mesenchyme cells, sitting at the tip of the archenteron and attaching filopodia to the blastocoel roof, and by planar intercalation of archenteron cells (*Hardin & Weliky, 2019*).

In *A. aurita*, secondary invagination starts at the mid-gastrula stage. It is based on the activity of the archenteron cells and the blastopore lip cells.

At the mid-gastrula stage, blastopore lip cells have very pronounced wedge shape, their apicobasal axes are skewed towards the oral pole (Figs. 3E, f; 4C and 4K). At the

mid-gastrula stage, apical surfaces of the blastopore lip cells become elongated towards the oral pole (Fig. 4G). These morphological features indicate that involution of the blastopore lip aids the archenteron invagination.

Involution of the blastopore lip was observed in the gastrulation of many model objects including amphibians (*Keller, Davidson & Shook, 2003*) and sea urchins (*Ettensohn, 2020*). However, the cellular mechanism of involution differs in different animals. In the gastrula of *Xenopus*, where the blastopore lip is several cell layers thick, involution occurs by the collective migration of mesodermal cells through the lip along arc-like trajectories (*Evren et al., 2014*). In *A. aurita*, the blastopore lip is an epithelial monolayer. We assume that the formation of bottle cells weakens the contact between the archenteron cells and blastopore lip cells. Therefore, the blastopore lip represents the monolayer with the free edge, which naturally curls at the free edge with the basal side inward (*Fouchard et al., 2020*). Rolling of the blastopore lip advances the archenteron deeper into the blastocoel, thereby contributing to the invagination of *A. aurita*.

Aborally directed migration of the archenteron cells is another mechanism that ensures the progression of the archenteron deep into the blastocoel. Note, the cell in Fig. 10E transforms its basal end into a typical leading edge with filopodia and lamellae characteristic of migratory cells. The leading edge with filopodia and lamellae is also characteristic of marginal bottle cells of the archenteron. These cells extend their filopodia towards the blastocoel roof and migrate along the ectoderm basal surface. They never leave the archenteron, which retains epithelial structure. Migrating cells generate the force that might be sufficient to drag the entire archenteron and the blastopore lip in the aboral direction. A similar mechanism of secondary invagination was described for another cnidarian species that gastrulates by invagination, *Nematostella vectensis*. By mathematical modeling, it was shown that the migratory activity of archenteron bottle cells contributes significantly to invagination of the archenteron in *Nematostella* (*Tamulonis et al., 2011*).

At the late gastrula stage, there are three types of embryo structure that differed in the morphology and behavior of the archenteron cells. In the first type (Figs. 5B, b; 9G1), marginal bottle cells migrate along the blastocoel roof, while the central cells are migratory passive (Fig. 10G). In embryos of the second type (Figs. 5F; 9G2), not only the marginal cells form the leading edge, but also the central cells (Fig. 10G). All the cells that can reach the blastocoel roof by their filopodia will migrate (Fig. 9G2). In embryos of the third type (Fig. 9G3), the archenteron central cells acquire the shape characteristic of migrating mesenchymal cells (Fig. 10E). They form many long filopodia at the leading edge and short protrusions at the trailing edge (Figs. 5D, 5G and 5H). It seems that archenteron cells in such embryos are connected to each other only by spot-like contacts between the protrusions formed at the leading and trailing edges. Almost all the archenteron cells migrate along the blastocoel roof (Fig. 9G3). We have noticed that this type of morphology is characteristic for embryos whose archenteron consists of a small number of cells. In such an embryo, the archenteron loses its integrity easily (Figs. 5E and 5H). The most aboral archenteron cells form a migration front, pulling the rest of the archenteron cells.

Anyway, in all embryos, the migrating archenteron cells maintain the contacts with neighboring archenteron cells and with the blastopore lip cells. Migration stops when

all the archenteron cells contact the the blastocoel roof (Figs. 6B and 6C; 9G1 and 9G2) or reach the aboral pole in the course of migration (Fig. 9G3). This event indicates the beginning of preplanula formation.

### How can we explain the contradictions in the data on Aurelia gastrulation?

According to most authors, gastrulation in *Aurelia* and other scyphozoans proceeds exclusively (or mainly) through invagination (*Goette, 1887*; *Hargitt & Hargitt, 1910*; *Yuan et al., 2008*). However, Hyde has found that gastrulation in *Aurelia* occurs by invagination and ingression (*Hyde, 1894*). In his excellent and accurate work, *Smith (1891)* comprehensively discussed the reasons for striking contradictions between the descriptions of *Aurelia's* development. He proposed several reasons for the misinterpretation of histological data on *Aurelia* gastrulation. In *Aurelia* gastrulae, the blastopore opening is so tiny that it could be easily overlooked on the thick paraffin sections, which were in practice in the XIX century. Oblique sections that were obtained by cutting inappropriately oriented embryos is another possible reason for interpretation of the *Aurelia* gastrulation as cell ingression or even delamination. An interesting point is also the presence of single cells in the blastocoel of *Aurelia* embryo (Figs. 2I and 2L; 3A) (*Hargitt & Hargitt, 1910*; *Hyde, 1894*; this study). Several authors suggested that these cells appear in the course of ingression and might participate in the endoderm formation (*Hyde, 1894*). We often found single cells in the blastocoel of *Aurelia* blastulae. These cells were variable in size and contained fragmented chromatin (Fig. 2L). Therefore, these cells necessarily degenerated at a later stage. Our data confirmed assumptions on the fate of these cells proposed by other authors (*Claus, 1883*; *Hargitt & Hargitt, 1910*; *Smith, 1891*).

### Segregation of the germ layers is linked to the blastopore healing

Gastrulation ends with the segregation of the germ layers. In *A. aurita*, segregation of the germ layers occurs when the closed blastopore heals during the preplanula stage (Figs. 7; 9H–9K). The successive stages of preplanula development have been already described by the methods of light microscopy (*Smith, 1891*; *Hargitt & Hargitt, 1910*). We confirmed and further detailed these findings using modern research techniques.

Soon after the archenteron cells have completely spread over the blastocoel roof, they begin to change shape. They reduce the protrusions on their former leading edges and acquire a cuboidal shape (Figs. 6B and 6C; 7C; 10G, g). A little later, the number of endodermal cells increases and cells acquire a more columnar shape (Figs. 7D; 10H, h). Cell reshaping is a morphological sign of epithelial endoderm formation.

Concurrently, the 'healing' of the blastopore, which closed at the late gastrula stage, starts (Figs. 7L–7N). In the late gastrulae / early preplanulae, the presumptive ectoderm and presumptive endoderm constitute a single continuous cell layer (Figs. 7C and 7L; 9H). The boundary between the germ layers can be detected by only analyzing cell morphology (Figs. 6D and 6E). Segregation of the germ layers is based on the fusion of the epithelial sheet edges, the process observed in embryonic development (*e.g.*, neural tube closure), and in wound healing (*Martin & Wood, 2002*). Soon after the closure of the blastopore opening (Figs. 6D; 7C), epithelial fusion starts in the presumptive endoderm, which completely separates from the blastopore lip and becomes a continuous layer (Figs. 7D–7F, 7L–7N;

9I). Then epithelial fusion starts in the outer cell layer. We assume that the cells of the former blastopore lip interdigitate. At first, cell interdigitation leads to the shortening of the inner part of the blastopore lip (the part internalized during invagination) (Figs. 7C–7F, 7H; 9H and 9I). Afterwards, the scar from the closed blastopore heals (compare Figs. 7H and 7J; 9J), and the ectoderm restores its integrity, which was lost at the beginning of germ layers segregation. That is the moment of final segregation of the germ layers and the end of gastrulation.

Therefore, in contrast to *Nematostella* development, the cells of the former blastopore lip integrate into the oral ectoderm of the animal pole of the embryo.

### Post-gastrulation events

Post-gastrulation development includes the formation of the basal lamina separating the germ layers, cell differentiation, further elongation of the oral-aboral axis, and morphological differentiation of the axis poles. Embryonic development ends with the formation of the competent planula larva, which is able to leave the medusa oral arms and initiate the metamorphosis (Fig. 8). Recently, anatomy and fine morphology of *Aurelia* planula were examined mostly by the methods of CLM (*Nakanishi et al., 2008*; *Yuan et al., 2008*). However, there is no detailed description of the planula cells' ultrastructure, and our TEM data (Figs. 8F–8N) reduce this gap.

In *Aurelia* larva, several cell types were found by the methods of CLM and TEM (*Nakanishi et al., 2008*; *Yuan et al., 2008*). The presence of nerve cells in the scyphozoan planula was confirmed by using anti-FMRFamide, antitaurine, and antityrosinated tubulin antibodies. It is worth mentioning that nerve cells in *Aurelia* planula were detected for the first time by histological method based on silver impregnation (*Korn, 1966*). Our study supports these findings (Figs. 8I–8K). Using the TEM, we detected the following cell types: ectodermal epithelio-muscular cells, endodermal epithelio-muscular cells, nerve cells, cnidocytes, and gland (secretory) cells (Figs. 8H–8Q).

The ectoderm in the competent planula has a pseudostratified structure (*Yuan et al., 2008*; this study). We found that the basally located nuclei in the presumptive ectoderm appear as early as at the mid-gastrula stage. At the late gastrula stage, it becomes evident that these nuclei belong to the small cells located at the base of the ectoderm in between the basal processes of other ectodermal cells (Figs. 6B and 6C). We agree with assumption that the small cells migrate to the base of the ectodermal layer (*Smith, 1891*) and differentiate into the cnidocytes (*Hargitt & Hargitt, 1910*), and possibly nerve cells.

It seems that the oralmost part of the planula ectoderm descend from the former blastopore lip cells (Figs. 7L–7N). It is known that the oral ectoderm of *A. aurita* planula gives rise to the ectodermal lining of the primary polyp manubrium (*Mayorova, Kosevich & Melekhova, 2012*). In the anthozoan *N. vectensis*, the primary polyp pharynx originates from the blastopore lip (*e.g.*, *Magie, Daly & Martindale, 2007*; *Steinmetz et al., 2017*). These data give us a cue for comparison between the planula metamorphosis and the primary polyp structure between anthozoan and scyphozoan cnidarians.

We have found that the endoderm of the competent planula is clearly regionalized. It can be subdivided into the three compartments, which differ in cell ultrastructure (Figs.

8E and 8G): the aboral (anterior) (Fig. 8O), the middle (Fig. 8P), and the oral (posterior) (Fig. 8Q). The ultrastructural differences might indicate the differences in the fate of these compartments during planula metamorphosis. Similar regionalization has been observed in the planula of several hydrozoans (*Burmistrova et al., 2018*; *Van de Vyver, 1964*; *Vetrova et al., 2021*), but the exact fate of each compartment remains unknown.

The hypothesis suggesting that the planula endoderm undergoes apoptosis during metamorphosis, and the polyp endoderm derives from the planula ectoderm (so-called "secondary" gastrulation), was proposed several years ago (*Gold et al., 2016*; *Yuan et al., 2008*). We do not intend to discuss this hypothesis here, as our study does not deal with the metamorphosis stage. Our data on the planula cells' ultrastructure and the developmental origin of the planula body parts can be viewed as an essential prerequisite for further study of the cellular mechanisms of *A. aurita* metamorphosis.

### How many ways do cnidarians use to invaginate? Comparison of invagination in A. aurita and Nematostella vectensis

Invagination is observed in the gastrulation of representatives of two cnidarian classes - Anthozoa and Scyphozoa. Among anthozoans, invagination was found only in the subclass Hexacorallia, whereas Octocorallia gastrulate by secondary (morular) delamination (*Kraus & Markov, 2017*). For scyphozoans, invagination seems to be a common way of gastrulation, along with cell ingression (*Berrill, 1949*). We cannot be precise about how common invagination is for scyphozoans, as embryogenesis has been studied only in about ten species. Embryos of different cnidarians that have already completed gastrulation by invagination look almost the same (Fig. 9H). The question arises how similar (or different) are the cellular mechanisms of invagination in the different species. In a broader context, it is a question of how different developmental pathways, leading to the formation of the same body plan, evolve. As the cellular mechanisms of cnidarian invagination have only been studied in the sea anemone *N. vectensis* (*Fritzenwanker, Saina & Technau, 2004*; *Kraus & Technau, 2006*; *Magie, Daly & Martindale, 2007*; *Pukhlyakova et al., 2019*; *Tamulonis et al., 2011*; for review see *Technau (2020)*) and the scyphozoan jellyfish *A. aurita* (this study), we will compare the invagination of these two species.

In *Aurelia* and *Nematostella*, invagination begins with the establishment of a domain whose cells contract the apices and gradually acquire a bottle shape (Figs. 3A–3J; 10B–10D; 11A–11D, 11H, 11I and 11K). At the later stages, these cells will constitute the archenteron (Figs. 11B, 11E and 11J) and form the larval endoderm. The formation of bottle cells is often associated with invagination of the epithelium and ingression of cells from the epithelium (*Pearl, Li & Green, 2017*; *Shook & Keller, 2003*). At the cellular level, the formation of bottle cells is a part of the epithelial-mesenchymal transition (EMT). During EMT, epithelial cells lose (or weaken) features of the epithelial phenotype and acquire ones of the mesenchymal phenotype (*Campbell & Casanova, 2016*; *Nieto et al., 2016*; *Shook & Keller, 2003*; *Yang et al., 2020*). So far, no morphological features or molecular markers of the mesenchymal state have been found that are universal for EMTs in all animals (*Yang et al., 2020*).

In species gastrulating by cell ingression (*e.g.*, in the hydrozoan *Clytia hemisphaerica*), presumptive endodermal cells acquire a bottle shape and ingress into the blastocoel

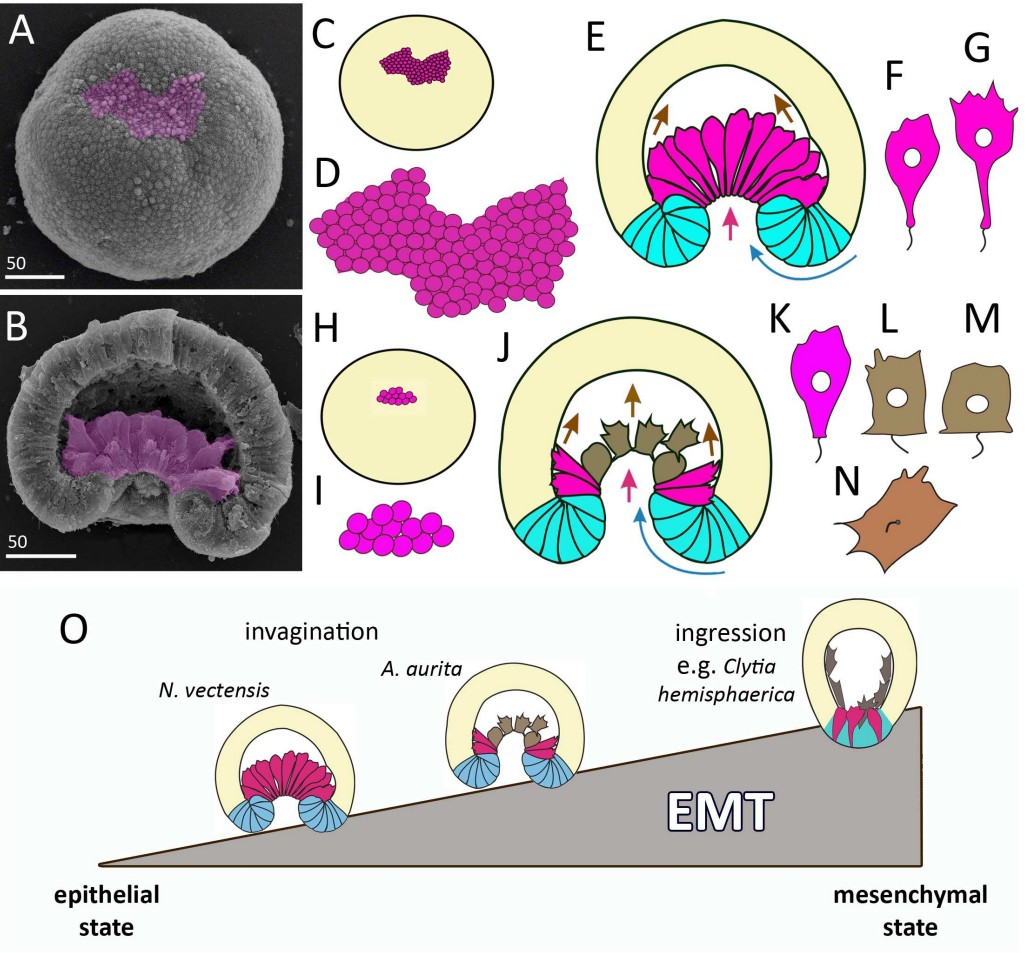

**Figure 11   Comparison of invagination during gastrulation in *Nematostella vectensis* and *Aurelia aurita*.** (A, B) SEM of *Nematostella* embryos. (A) *Nematostella* early gastrula, very beginning of archenteron invagination; preendodermal plate consisting of presumptive archenteron (endoderm) cells is artificially colored in magenta. (B) *Nematostella* mid-gastrula split into halves. Bottle cells of the archenteron are colored magenta. (C–G) Schematic representation of *Nematostella* gastrulation. Early gastrula (C) and preendodermal plate consisting of about 200–300 cells (D). (E) Scheme of *Nematostella* mid-gastrula. (F, G) Bottle cells of *Nematostella* archenteron. (H–M) Schematic representation of *Aurelia* gastrulation. Early gastrula (H) with the archenteron area consisting of about 20 cells (I). (J) Scheme of *Aurelia* mid-gastrula; archenteron consists of morphologically different cells (K–N). (O) EMT in cnidarians gastrulating by invagination and ingression is represented as a gradient of cell states between two extremes—epithelial and mesenchymal phenotypes. Red arrows—invagination; blue arrows—involution of the blastopore lip; brown arrows—migration of the archenteron cells towards the aboral pole.

individually (*Kraus, Chevalier & Houliston, 2020*). In contrast, in species that gastrulate by invagination, presumptive endodermal cells are organized into a single oral domain, do not alternate with columnar epithelial cells, and pass through the main stages of EMT almost synchronously (Figs. 10B–10D).

The number of presumptive endodermal cells in the oral domain is much lower in *A. aurita* than in *N. vectensis*. At the early gastrula stage, there are about 250 bottle cells

(237 $\pm$ 32; N embryos = 10) in *N. vectensis*, while in *A. aurita*, there are about 20 bottle cells only (17.5 $\pm$ 4.5; N embryos = 10) (Figs. 11C, 11D, 11H and 11I). The number of endodermal cells in *A. aurita* increases only during the transition from the late gastrula to the competent planula stage.

The apical constriction in the oral domain cells of *Aurelia* is not synchronized from the very beginning. The same phenomenon was observed in *N. vectensis* (*Kraus & Technau, 2006*). Interestingly, the synchronous and asynchronous phases in the contraction of cells' apices are also observed during mesoderm invagination in *Drosophila* (*Oda & Tsukita, 2001*).

The various stages of EMT do not show a conserved sequence nor necessarily proceed to the final stage (*Campbell & Casanova, 2016*; *Nieto et al., 2016*; *Shook & Keller, 2003*). We observed the same in the EMTs of *Aurelia* and *Nematostella*. At the beginning of gastrulation, EMT proceeds through the classical stages of bottle cells formation (*Shook & Keller, 2003*): cells elongate apico-basal axes and slightly constrict apical perimeters; then cells sharply constrict apices, extend basal ends and shorten apicobasal axes (Figs. 10A–10D) (*Kraus & Technau, 2006*; this study). Interestingly, bottle cells of both species do not lose cilia (Figs. 3G and 3K; 11F and 11K). However, the stages that the bottle cells pass through later in development are different between *A. aurita* and *Nematostella* (Figs. 11F, 11G and 11K–11N).

During EMT, the cell do not always reaches the extreme (mesenchymal) state. The behavior of cells undergoing EMT may vary from individual migration to collective migration as a cohort, in which cells maintain intercellular contacts (*Campbell & Casanova, 2016*). EMT can be viewed as a continuum of cell states, with epithelial and mesenchymal phenotypes being the extremes (*Nieto et al., 2016*).

In cnidarians gastrulating by cell ingression (*e.g.*, in the hydrozoan *Clytia hemisphaerica*), presumptive endodermal cells acquire a bottle shape and ingress into the blastocoel individually as mesenchymal cells (Fig. 11O) (*Kraus, Chevalier & Houliston, 2020*). In contrast, we consider that EMT remains incomplete in cnidarians that gastrulate by invagination (see *Technau, 2020*). In *Nematostella*, EMT arrests far prior to its final stage (*Magie, Daly & Martindale, 2007*) (Fig. 11O). The bottle cells further elongate their apicobasal axes (Figs. 11F and 11G). These cells form typical leading edges with lamellae and filopodia (*Kraus & Technau, 2006*). The cells closest to the blastocoel wall start migrating along it towards the aboral pole (Fig. 11E) (*Tamulonis et al., 2011*). The bottle cells retain subapical contacts connecting them with each other and with the cells of the blastopore lip (*Kraus & Technau, 2006*). Migratory behavior affects the shape of the bottle cells: the formation of a long thin neck (Fig. 11G) is due to the fact that the cells move aborally, remaining connected with each other. The cells retain their bottle shape until the late gastrula stage when they get into contact with the aboral pole ectoderm.

In *Aurelia*, by the mid-gastrula stage, most presumptive endodermal cells lose the bottle shape (Figs. 4E, e, 4H, h, 4I, 4J, 4M; 10E–10G). Only cells at the base of the archenteron retain a bottle shape and migrate along the blastocoel wall like *Nematostella* cells (compare magenta cells in Figs. 11E and 11J). In *Aurelia*, the archenteron cells are much closer to the mesenchymal phenotype than in *Nematostella* (Fig. 11O). Many archenteron cells

acquire a state of collectively migrating cells linked to their neighbors by spot-like contacts. Indeed, most aboral cells form a migratory front pulling other archenteron cells towards the aboral pole (Figs. 5E; 9G2 and 9G3). In *Nematostella*, archenteron cells migrating towards the aboral pole do not lose the bottle shape and subapical contacts. In both species, the coherent behavior of archenteron cells resembles that of the cells migrating during the wound healing (*Du Roure et al., 2005*; *Poujade et al., 2007*).

We assume that *A. aurita* and *N. vectensis* might differ in the relative contribution of primary invagination based on the cell apical constriction and involution of the blastopore lip to gastrulation (Figs. 11E and 11J). It can be explained by the difference in the number of presumptive endodermal cells between *A. aurita* and *N. vectensis*: 200 cells (in *N. vectensis*) generate a higher traction force than 20 cells (in *A. aurita*) (Figs. 11C, 11D, 11H and 11I). That is why the involution of the blastopore lip might have a higher impact on the sinking of the archenteron into the blastocoel in *A. aurita* than in *N. vectensis*.

## CONCLUSIONS

In this study, we aimed to uncover the mechanisms underlying *A. aurita* early development at cell- and tissue- levels. This species exhibits a canonical cnidarian cleavage, which is associated with the formation of a coeloblastula, gastrulation via invagination, and development of a planula larva with the completely closed blastopore typical for hydrozoan and scyphozoan species. We reconstructed the sequence of events characteristic of the *A. aurita* gastrulation as follows. Gastrulation starts from the morphological differentiation of the oral domain of columnar cells at the pregastrula stage. In early gastrula stage embryos, primary invagination occurs as a bending of the oral epithelium based on changes in the shape of its cells. These cells acquire a bottle shape by constricting their apices and enlarging their basal ends. At the mid-gastrula stage, secondary invagination, which brings the archenteron to its final position, begins. In *A. aurita*, secondary invagination is based on the migratory activity of archenteron cells and involution (rolling) of the blastopore lip. At the late gastrula stage, blastopore closure occurs, marking the end of gastrulation, and the blastopore lip cells become incorporated into the oral ectoderm.

Through comparative analysis of *Aurelia* and *Nematostella* gastrulation, we clearly show that invagination differs significantly at the cellular level between species belonging to the same phylum, but phylogenetically distant from each other. The differences primarily concern the cells of the archenteron. The number of cells involved in invagination, the dynamics of cell shaping, the stage of EMT that the cells reach, and the variety of forms and behaviours of archenteron cells within the same embryo may differ. Differences also lie in the relative roles of primary invagination, secondary invagination and involution of the blastopore lip in the dipping of the future endoderm into the blastocoel.

Thus, in *Aurelia*, much fewer cells are involved in invagination. Its secondary invagination relies more on involution than that of *Nematostella*. In contrast to archenteron cells of *Nematostella*, the *Aurelia's* cells lose their bottle shape very quickly, and their morphology and behaviour are very diverse within the same embryo. We suggest that *Aurelia'* endoderm cells progress further in the EMT than *Nematostella'* cells. However,

they never reach the state of individually migrating mesenchymal cells characteristic of cnidarians that gastrulate by unipolar ingression (*e.g.*, *C. hemisphaerica*).

Further comparative studies on cnidarians that gastrulate by invagination will clarify whether a similar level of difference is observed when comparing invaginations of species not so distant from each other as *Aurelia* and *Nematostella*.

## ACKNOWLEDGEMENTS

We like to acknowledge the staff of N.F. Pertzov White Sea Biological Station of Lomonosov Moscow State University, Russia, for providing opportunity for the research and equipment usage of the Center of microscopy WSBS. Electron microscopy was carried out at the Shared Research Facility ''Electron microscopy in life sciences'' at Moscow State University (Unique Equipment ''Three-dimensional electron microscopy and spectroscopy''). We thank Stanislav Kremnyov for the CLM image used for Fig. 6D.

### Funding

The research was supported by the Russian Foundation for Basic Research (grant # 19-04-01131a (Igor Kosevich)), and was carried as part of the Scientific Project of the State Order of the Government of Russian Federation to Lomonosov Moscow State University #121032300118-0 (Boris Osadchenko, Igor Kosevich) and Governmental Basic Research Program for the Koltzov Institute of Developmental Biology of the Russian Academy of Sciences #0088-2021-0009 (Yulia Kraus). The funders had no role in study design, data collection and analysis, decision to publish, or preparation of the manuscript.

### Grant Disclosures

The following grant information was disclosed by the authors:
Russian Foundation for Basic Research: #19-04-01131a.
Scientific Project of the State Order of the Government of Russian Federation to Lomonosov Moscow State University: #121032300118-0.
Governmental Basic Research Program for the Koltzov Institute of Developmental Biology of the Russian Academy of Sciences: #0088-2021-0009.

### Competing Interests

The authors declare there are no competing interests.

### Author Contributions

- Yulia Kraus conceived and designed the experiments, performed the experiments, analyzed the data, prepared figures and/or tables, authored or reviewed drafts of the paper, and approved the final draft.
- Boris Osadchenko performed the experiments, analyzed the data, authored or reviewed drafts of the paper, and approved the final draft.

- Igor Kosevich conceived and designed the experiments, performed the experiments, analyzed the data, prepared figures and/or tables, authored or reviewed drafts of the paper, and approved the final draft.

## Data Availability

The raw measurements are available in the Supplementary Files.

## Supplemental Information

Supplemental information for this article can be found online at http://dx.doi.org/10.7717/peerj.13361#supplemental-information.

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
