# Peer review of "Embryonic development of the moon jellyfish Aurelia aurita (Cnidaria, Scyphozoa): another variant on the theme of invagination"

_PeerJ, doi:10.7717/peerj.13361_

## Round 0.1 · original submission · Major Revisions

Dear colleagues, I ask you to carefully read the comments of the reviewers, make the appropriate corrections and send the corrected version to the journal.

Reviewer 1 ·

Basic reporting

This paper aims at defining evolutionary pathways based on the development of the early lifre stages of the scyphozoan Aurelia aurita by comparing them with the anthozoan Nematostella vectensis.
To achieve this goal, the authors provide data and images realised with the advanced microscopy techniques which provide a better description compared to the previous available literature (e.g. Berrill 1949) and highlight patterns remained little or not known at all to present.
Overall, the paper is interesting, clear in the aim and supported by data.

Experimental design

appropriate

Validity of the findings

Despite theoverall quality, the paper is too long in my opinion, so that it ends up being a very long description and the goal to define the difference with Nematostella and anthozoans in general gets lost in the words. Actually, the Discussion appears to be a mixture of Results and Discussion. The Results are already too long and detailed. In my opinion, the authors should make an effort to shorten the whole paper. For example, in the M & M there are two paragraphs at the beginning which do not add anything new, and could therefore be extremely reduced in lenght (please see Detailed comments below). My suggestion to the authors is to be detailed when they highlight new patterns, but go very fast if these patterns were already observed in previous studies. The conclusons are not conclusive at all, they are just what the authors hope, but they do not synthetize the findings of the study.
Finally, despite the fact that the use of scientific English is overall correct, I have found here and there some flaws (see Detailed Comments below). I could not report them all, but I suggest the authoors to critically revise the paper and pay attention to them.

Additional comments

Deatailed Comments
L26: I think metazoan’s should be metazoans’
L 33: I suggest the authors remove “molecular”: they do not use a molecular approach, so this word is confusing the reader
L64-70: I think here you need to add references, although these topics may be known by a large audience. In particular, I think the classification into Medusozoa may not be known by non-specialist in the field
L 75-77: the authors appear to contradict themselves in this sentence…additionally, I disagree with the statement that Aurelia is difficult to breed in captivity, considered that it is one oof the first and stilli s very coommonly exhibited in public Aquaria because ot is easy to breed in captivity throughout the whole life cycle
L78-79: I disagree with this statement, as Fuchs et al. themselves highlighted in their paper that the role played by retinoic acid during polydisk strobilation is likely only a first step toward the understanding of the moledular signaling behind strobilation and several questions are left open to future research
L86-87: it may be simply my own preference, but I would rephrase: “It may seem that we already know everything about Aurelia early development” because it seems more appropriate to a divulgative text than a rigorous scientific paper
L87-95: the aim of this paper comes out of the blue here…why it is so important to understand the difference(s) of development in Scyphozoa and Anthozoa? Why was Nematostella chosen for the comparison? Here as well as in the background, there is the word “molecular”: the authors should precise that they do not use this approach, because this generates confusion
L109-123: As these aspects were already described, these paragraphs appear redundant to me…in my opinuion, they could be extremely shortebed abd added to the last paragraph
L620: I think “How many ways use cnidarians to invaginate?” should be “How many ways do cnidarians use to invaginate?”

·

Basic reporting

This manuscript from Kraus et al. describes at the cellular level the developmental stages of the scyphozoan Aurelia aurita from zygote to planula stage, using a combination of light, confocal and electron microscopy. The manuscript is well written and cite the relevant literature. Raw data are shared.

Experimental design

The Authors describe in great details the cellular dynamics during gastrulation and show that the formation of the endodermal layer involves a combination of invagination, EMT and cell rearrangement.

Validity of the findings

The quality of the data is excellent, the figures beautifully composed and the description and discussion of the data comprehensive and insightful. This descriptive work will certainly represent a milestone in scyphozoan embryology.

Additional comments

I would suggest a few corrections before publication:
L25, L64. I would avoid using the term "archetypal" in biology. Could be replaced by “emblematic” for instance.
L32. “occurring during gastrulation” instead of “in the gastrulation”.
L34, 36, 46, 47, 54, 109, 197, as well as the references. Please put species names in italic.
L55-56. The sentence starting with “We clearly show that….” is too generic. You could either detail the observed differences between species or delete this sentence.
L56 “between species” instead of “between the species”.
L103. “might be based on very different cellular mechanisms” this seems an overstatement. Here I would suggest replacing this part by a quick description of the main differences between Nematostella and Aurelia gastrulation.
L109-123. This section does not describe the studied materials or used methods. I would suggest transferring most of it at the beginning of the Results section.
L120. the sentence “and never…. in the ‘gonads" is difficult to read. Please rephrase.
L137. What is the composition and concentration of the “phosphate buffer”.
L330 and L560. It is not mentioned (L330) or discussed (L560) what the blastopore cells become during the final steps of blastopore closure – are they integrated in the endodermal and/or ectodermal layers? Please discuss this matter.
L430. “blastomeres resembling the same during spiral or radial cleavage in high metazoans”. This sentence is confusing should be rephrased or removed. The use of “high metazoans” should be avoided.
L463. Please correct “in the the gastrula”.
L540. “we almost confirmed the data”. Why “almost”? Please explained.
L577. It is mentioned that nerve cells were identified by TEM – but I could not find any mention of this in the Results section or in the figures. Please provide images and describe the presence of nerve cells at the planula stage.
L691. “the cell far from always” – a word is missing here.
Legend of Figure 1. “the blastomeres exhibit packing reminiscent of spiral cleavage”. Here again the parallel with spiral cleavage seems an overstatement. I would suggest deleting it.
Figure 2b. I would suggest coloring the right part of the blastopore also in light blue.

---

## Round 0.2 · accepted · Accept

Dear Dr. Kosevich, thank you for your meticulous work on improving the manuscript.